



# Influence of radiative forcing factors on ground-air temperature coupling during the last millennium: implications for borehole climatology

Camilo Melo-Aguilar [1,2], J. Fidel González-Rouco [1,2], Elena García-Bustamante [3], Jorge Navarro-Montesinos [3], and Norman Steinert [1,2]

[1]Universidad Complutense de Madrid, 28040 Madrid, Spain
[2]Instituto de Geociencias, Consejo Superior de Investigaciones Cientificas-Universidad Complutense de Madrid, 28040 Madrid, Spain
[3]Centro de Investigaciones Energéticas, Medioambientales y Tecnológicas (CIEMAT), 28040 Madrid, Spain

**Correspondence:** Camilo Melo-Aguilar (camelo@ucm.es)

**Abstract.** Past climate variations may be known from reconstruction methods that use proxy data as predictors. Among them, borehole reconstructions is a well established technique to recover the long-term past surface air temperature (SAT) evolution. It is based on the assumption that SAT changes are strongly coupled to ground surface temperature (GST) changes and transferred to the subsurface by thermal conduction. We evaluate the SAT-GST coupling during the last millennium (LM) using simulations

from the Community Earth System Model LM Ensemble (CESM-LME). The validity of such premise is explored by analyzing the structure of the SAT-GST covariance during the LM and also by investigating the evolution of the long-term SAT-GST relationship. The multiple and single-forcing simulations in the CESM-LME are used to analyze the SAT-GST relationship within different regions and spatial scales and derive the influence of the different forcing factors on producing feedbacks mechanisms that alter the energy balance at the surface. The results indicate that SAT-GST coupling is strong at global and

above multi-decadal time scales in the CESM-LME however a relative small variation in the long-term SAT-GST relationship is also represented. Although at global scale such variation does not impact significantly the SAT-GST coupling, at local to regional scales this relationship experiences considerable long-term changes mostly after the end of the 19th century. Land use land cover (LULC) changes are the main driver for decoupling SAT and GST locally and regionally since they modify the land surface properties such as albedo, surface roughness and hydrology, and thus the energy fluxes at the surface. Snow cover

feedbacks due to the influence of other external forcing are also important for corrupting the long-term SAT-GST coupling. Our findings suggest that such local and regional SAT-GST decoupling processes may represent a source of bias for SAT reconstructions from borehole measurement since the thermal signature imprinted in the subsurface over the affected regions is not fully representative of the long-term SAT variations.

## 1   Introduction

Improving our knowledge of the LM climate variability is key for a better understanding of the mechanisms that determine the Earth System response to natural (solar, volcanic and orbital) and anthropogenic (land use and atmospheric composition



changes) external forcings (Masson-Delmotte et al., 2013). Instrumental records represent the most adequate alternative to study past climate variations. However, they only provide coverage since the mid of the 19th century (e.g. Hansen et al., 2010; Jones et al., 2012). Therefore to understand the nature of climate variability operating on longer temporal scales, LM reconstructions from a variety of proxy data (e.g. tree-rings, corals, preserved pollen, ice cores, etc.; Jones et al., 2009) and
simulations using general climate models (GCMs) are generally employed.

The current available LM temperature reconstructions agree in depicting the general pattern of the temperature evolution going from a relatively warmer period at the beginning of the LM (MCA; Medieval Climate Anomaly) to a colder period from about 1450 to ca. 1850 (LIA; Little Ice Age), interrupted by the industrial warming in the 19th century (Masson-Delmotte et al., 2013). Despite this general agreement there is still a large range of uncertainty that stems from different sources, in-
cluding: different reconstruction methods; various calibration and verification processes; spatial and temporal coverage; the different proxy locations (land, land and ocean, etc.) and the alternative statistical methods employed (e.g. Jones et al., 2009; Fernández-Donado et al., 2013). For instance, the range of uncertainty during the MCA in reconstructing Northern hemisphere (NH) temperature is of about 0.6°C. Also, Ammann and Wahl (2007) estimate that the cooling during the Maunder Minimum (1645-1745) relative to present is about 0.7°C while Christiansen and Ljungqvist (2012) report a colder LIA (1.4°C) similarly
to findings by Pollack and Smerdon (2004).

Addressing the range of uncertainties in reconstructing past temperature changes is relevant not only for assessing our understanding of past temperature changes and the confidence we have on available estimates (Masson-Delmotte et al., 2013), but also for model-data comparison exercises (e.g. Fernández-Donado et al., 2013) and for constraining the range of estimates of the system response to changes in the external forcing, i.e. the sensitivity of the climate system (Hegerl et al., 2006; Fernández-
Donado et al., 2013)

Borehole temperature inversion is a well established reconstruction technique and leans on two main assumptions: First, surface air temperatures (SAT) are expected to be closely coupled to ground surface temperatures (GST); The second, postulate is that variations in SAT propagate downward to the subsurface through conduction (Pollack et al., 1998; Smerdon et al., 2003, 2004). As a result, a thermal signature of the surface temperature is imprinted in the subsurface. This reconstruction technique
is limited to recovering only the low frequency information (decadal, and longer time-scales) since the soil acts as a low-pass filter and progressively filters out the higher-frequency variations with depth. Oscillations with periods on the order of days penetrate only ~50 cm deep (Smerdon and Stieglitz, 2006), seasonal cycles are solely observed close to the surface (<10 m depth), decadal variations propagate within the upper 50 m and multi-centennial changes within the LM are observed in the upper 500 m below the surface (Mareschal and Beltrami, 1992; Pollack and Huang, 2000).

As every type of reconstruction method, the borehole technique is also subject to diverse sources of uncertainties. One of them is that the assumption of a conductive heat transfer may not always be substantiated, since non-conductive processes such as advection and convection may influence or even dominate the subsurface thermal regime (Kukkonen and Clauser, 1994) in areas with important groundwater flows or geothermal activity. Nevertheless, a number of studies suggest that the impacts of such processes can be reduced with an appropriate treatment of the affected borehole temperature logs (e.g Kohl, 1998; Bodri
and Cermak, 2005; Ferguson et al., 2006)



Additionally, the most important contributions to uncertainty arise from the impacts of changes in surface processes that affect the SAT-GST coupling. Snow cover is especially important since it insulates the ground surface from the cold winter air causing large differences between the soil and air temperatures (e.g. Pollack and Huang, 2000; Stieglitz et al., 2003; Beltrami and Kellman, 2003; Bartlett et al., 2004). The impact of this effect on borehole theory has caused considerable discussion. Mann

and Schmidt (2003) argued that borehole-based reconstructions may be substantially biased by this seasonal influence of snow cover. Hence, under changing snow cover conditions GST may be not completely representative of SAT variations. Chapman et al. (2004) argued against based on the assumption that SAT and GST coupling is strong at longer than seasonal timescales, therefore snow biases would influence only the high frequency oscillations. Additionally, the long-term SAT-GST coupling has been also supported by both observations (Smerdon et al., 2003, 2004) and modeling assessments (González-Rouco et al.,

2003, 2006, 2009; Bartlett et al., 2005). Besides snow cover, other land surface and soil properties such as soil water content, vegetation and land use land cover (LULC) changes have also the potential of impacting the SAT-GST coupling. For instance, deforestation, afforestation and other land cover changes modify the land-surface properties including albedo, roughness, and evapotranspiration altering the energy transfer between the atmosphere and the ground (Anderson et al., 2011). In a recent study, MacDougall and Beltrami (2017) found that deforestation tends to warm the ground surface mainly by reducing the

transport of heat away from the surface. They found that such continuous vegetation changes would drive to long-term surface temperature anomalies and thus, deforestation should be considered as a possible source of bias for temperature reconstructions from subsurface temperatures.

One way of addressing uncertainties of paleoclimate reconstruction methods is using climate model simulations as a surrogate reality in which pseudoproxy records of varying complexity are created and reconstruction methods are replicated in pseudo

proxy experiments (PPEs) mimicking real world cases (Smerdon, 2012). The robustness of using the borehole method for reconstructing past SAT variations has been tested in PPEs (e.g. González-Rouco et al., 2006, 2003, 2009; Beltrami et al., 2006). González-Rouco et al. (2003) investigated the relationship between simulated SAT and GST from interannual to centennial timescales in a forced climate simulation of the LM (1000-1900 common era; CE) with the ECHO-G GCM (Legutke and Voss, 1999). They found that in spite of the seasonal and longer-term variability of snow cover, the coupling was stable

for decadal and above timescales, so that GST should be a good proxy for long-term SAT variations. González-Rouco et al. (2006) extended this analysis by implementing the borehole method in the simplified reality of the same model. They simulated underground temperature perturbation profiles using a heat-conduction forward model driven by simulated GST. Then, they applied an inversion approach to reconstruct ground surface temperature histories from the simulated profiles. Their results supported the overall performance of the borehole methodology.

Likewise, García-García et al. (2016) employed a similar approach as in González-Rouco et al. (2006) to retrieve past GST variations from GCMs applying an inversion method as well. They simulated global synthetic temperature profiles using a one-dimensional conductive model driven by simulated GST as an upper boundary condition from the CMIP5/PMIP3 (Coupled Model Intercomparison Project Phase 5 - Paleoclimate Modeling Intercomparison Project Phase 3; Taylor et al., 2012) LM simulations. Unlike the GCMs used in the early works that do not incorporate some external forcings, the PMIP3/CMIP5 sim-

ulations include a larger representation of LM forcings than pre-PMIP3 experiments (Fernández-Donado et al., 2013; Taylor



et al., 2012) including a variety of land-surface model components. Their results reinforced the reliability of recovering past global surface temperature variations from subsurface temperature measurements in PPEs experiments using current state-of-the-art GCMs.

The previous analyses support estimations of global/hemispheric past temperature obtained from borehole temperature inversions. First, they support the overall performance of the methodology in retrieving past GST histories from borehole profiles. Second, the use of up-to-date forcing representations in CMIP5 model ensembles also ensures that long-term alterations of surface properties like those induced by LULC changes, the effect of anthropogenic aerosols cooling and the potential long-term snow cover feedbacks induced by both forcings, do not seem to bias inversion results at global/hemispheric scales. In spite of these positive results, the analysis of last generation model experiments including the complete set of agreed CMIP5 forcings (Schmidt et al., 2011, 2012) in the so-called all-forcing experiments does not allow for insights into the individual effect of each of the forcing and the related feedbacks, both at global/hemispherical or at continental/regional scales. This would be desirable in order to quantify the effect of the individual forcings on SAT-GST coupling, and to obtain estimates of its particular temporal evolution that would allow for disentangling its contribution from that of other forcings. Therefore, the present work considers all-forcings and single-forcing types of experiments in order to address how state-of-the-art climate models simulate SAT-GST coupling from global to regional scales and to evaluate the potential influence of the external forcings on the SAT-GST relationship. This will also provide information about where and when a decoupling of SAT-GST may exist, with implications for borehole inversion practices at different spatio-temporal scales. For this purpose we use the Community Earth System Model-Last Millennium Ensemble (CESM-LME; Otto-Bliesner et al., 2016), which to date is the largest existing ensemble of LM simulations with a single model and has not been used in previous assessments of this kind. The CESM-LME includes all- and single-forcing experiments that consider jointly or separately the transient evolution of solar variability, volcanic activity, orbital changes, GHGs, anthropogenic aerosols and LULC (see details in Sect. 2)

Therefore, using all-forcing simulations (ALL-F hereafter) allows for evaluating the SAT-GST relationship throughout the LM in a realistic representation of the real world conditions. Additionally, the single-forcing experiments are suitable to identify the specific role that each forcing might play and its 'fingerprint' (Hegerl et al., 2011) on the presence of biases in the SAT-GST coupling. Most external natural (solar and volcanic variability) and anthropogenic (GHGs, LULC and aerosols) have the potential to indirectly affect SAT-GST thermodynamics through snow cover feedbacks (Hartmann et al., 2013). Additionally, LULC changes can also have direct influence in SAT-GST. During the LM the Earth's land cover has been substantially modified by the replacement of natural ecosystems with agricultural land, especially since the industrial period (Pontgratz et al., 2008; Hurtt et al., 2009). Such land use changes have the potential of altering the surface energy balance by the modification of energy fluxes and moisture budgets (Betts, 2001; Betts et al., 2007; Brovkin et al., 2006; Anderson et al., 2011; Zhao and Jackson, 2014) driving to direct impacts on the atmosphere and ground-surface temperature relationship (MacDougall and Beltrami, 2017). Furthermore, vegetation changes may additionally lead to indirect non-linear effects that are relevant in the borehole context. The most important one is the fact that deforestation at high latitudes leads to an increase in snow cover since low vegetation accumulates continuous snow cover more readily than forests in early winter favoring its permanence longer in spring (Myhre et al., 2013). The local, regional and large scale implications of these interactions on long-term climate variability and




specifically on the validity of the borehole method assumptions have not been explored so far.

The first part of the manuscript (Sect. 2) describes the CESM-LME simulations and forcings used in the experiment. Subsequently, Sect. 3 describes the methodological approach employed for the analysis of the coupling between SAT and GST during the LM. Section 4 presents the main results in the analysis of the global covariance structure of SAT, GST and soil temperature

(ST) at different model depths throughout the LM. This analysis includes the global SAT-GST long-term coupling at annual and seasonal timescales. The latter helps identifying the impacts of seasonality on the SAT-GST coupling. In addition, the spatial distribution of the covariance structure as well as the SAT-GST offset are also illustrated. This allows for the detection of possible failures in the air/ground temperature coupling at regional and local scales. The spatial analysis is extended to investigate the heat transfer within the shallow subsurface by comparing GST relative to deep model layers. Section 4.1 specifically

addresses the long-term trend of the SAT-GST relationship in order to evaluate whether this association experiences variation with time during the simulated LM. Finally, Sect. 5 provides a discussion about the implications of decoupling processes at different temporal and spatial scales for the borehole temperature reconstruction method.

## 2 Simulations and forcings

This analysis considers LM simulations produced with version 1.1 of CESM (CESM1; Hurrell et al., 2013). The Community Atmosphere Model version 5 (Neale et al., 2012) is used as the atmospheric component, the Parallel Ocean Program version 2 (Smith et al., 2010) represents the ocean component, including the Los Alamos sea ice model (Hunke et al., 2015). The horizontal resolution of the CESM-LME is ~2° over the atmosphere and land and ~1° in the ocean and sea ice (Otto-Bliesner et al., 2016).

The land surface component in the CESM1 is the Community Land Model Version 4 (CLM4; Lawrence et al., 2011) which incorporates some improvements relative to the previous version (CLM 3.5; Oleson et al., 2008) in the representation of land surface processes that are important for the energy transfer between the atmosphere and the soil (Oleson et al., 2010). Some of these include a better description of ground evaporation, thermal and hydrologic properties of organic soil, the ground depth, snow albedo, snow cover fraction and burial fraction of vegetation by snow (Lawrence et al., 2011). In addition, The CLM4

has the deepest bottom boundary condition placement (BBCP) among the current land surface models to date, placed at 42.1 m depth, discretized into fifteen layers (see Table 1) and including up to five additional layers in the overlying snow pack. This is relevant for borehole reconstruction applications because previous works have shown that shallow BBCs may not represent reliably the downward propagation of the temperature signal corrupting the amplitude attenuation and the phase shift (Smerdon and Stieglitz, 2006; Nicolsky et al., 2007; Alexeev et al., 2007), particularly for long-term trends as decadal from the compari-

son of energy storage in model simulations and borehole profiles (e.g. MacDougall et al., 2010). Therefore, this improved and deeper land surface model allows for analyzing the coupling between SAT and GST in a more realistic atmosphere-subsurface heat transfer scheme.

The CESM-LME incorporates ALL-F simulations with natural and anthropogenic forcings that were chosen following those



used in Landrum et al. (2012). It is composed by a total of 30 simulations including a subset of 10 simulations that incorporate all the LM external forcings and smaller subsets of single-forcing simulations considering each forcing individually (see Table 2 for details and references therein). The simulations used in this study and how they trace back to the original experiment names produced by Otto-Bliesner et al. (2016) are shown in Table 3.

## 3 Methods

In the present work the LM refers to the period from 850 to 2005 CE while the periods from 850 to 1850 and 1851 to 2005 CE refer to pre-industrial and industrial, respectively. 2 m air temperature is used for SAT and the first land model level (Table 1) represents GST. Additionally, the ST at different depths are used for addressing different aspects of the subsurface heat

transport within the CESM-LME. The simulated ST does not include data over the Antarctic region therefore this region is excluded from the analysis.

The relationship between SAT and GST is analyzed from two complementary perspectives. First, the covariance structure during the LM of the global SAT and GST anomalies relative to 1851-2005 CE is assessed at interannual scale. Likewise, the spatial patterns of the correlation between SAT and GST are also analyzed. This spatial analysis is extended to the differences

between SAT and GST LM mean values as this gives an additional measure of the energy exchange across the air-ground interface (Bartlett et al., 2004) that may inform about a potential air-ground temperature decoupling. In addition, the same spatial analysis is done for the differences between GST and $ST_{L8}$ in order to get insights into processes affecting the pure conductive subsurface transport assumption within the shallow subsurface. For specific cases where SAT and GST exhibit some kind of decoupling, a description of the main processes that lead to this decoupling at local and regional scales is presented. For this

purpose, the evolution of SAT, GST and ST at different depths is assessed during the last 105 years of the simulation period (1900-2005 CE) for the cases of interest. In some particular examples additional variables (e.g. latent and sensible heat fluxes or snow cover) are also included for a better description of the physical processes. The purpose of such analyses is providing some examples rather than detailing the minutiae of the processes that lead to different SAT-GST responses at many different locations. For the analysis described above the ALL-F ensemble will be used, thus, providing from the point of view of the

forcings considered a more complete and realistic representation of real world conditions than the single forcing runs. This allows for using the surrogate reality of the CEMS-LME model as a test bed for detecting potential sources of deviations in the SAT-GST relationship during the LM.

Second, we focus on the assessment of the long-term trend in the SAT-GST relationship. Non-stationarity in such association at long timescales (multi-decadal to multi-centennial) might imply that GST variations would not be representative of the SAT

variations, thus rendering unreliable inferences of past climate change (Bartlett et al., 2004). A two-phase regression model (Solow, 1987) is applied to address this issue, that allows for detecting changes in the long-term trend of the SAT-GST relationship. The year-of-change is identified, as well as the trend before and after the change. The timing of change and the magnitude of trends are suggestive of long-term changes in surface coupling. Initially, this assessment is performed at a global scale by





analyzing the LM evolution of global SAT minus GST anomalies. The analysis of the long-term trend is then extended to the spatial domain starting from the evaluation of SAT and GST linear trends independently during the industrial period since the largest changes are expected to occur within this period. In this part besides the ALL-F ensemble we have also considered the GHG-only, LULC-only and OZ/AER-only ensembles because they have the highest potential of altering the land surface

characteristics. This allows for identifying the influence of each forcing on the LM SAT-GST coupling, including changes in snow cover, soil moisture, latent and sensible heat variations. The linear trend analysis of SAT and GST gives a general view of their evolution through the industrial times and a first glance at the most important areas where long-term SAT-GST decoupling may exist. Finally, considering that the timing of variations in the land surface properties may have started before the industrial period, the two-phase regression analysis is also applied to the complete LM.

## 4 Results

CESM-LME supports the assumption that SAT is tightly coupled with GST at global scales and above multi-decadal scales. Figure 1a illustrates the stability of the coupling at global scales during the LM in the ALL-F$_2$ ensemble member as an example. Results are comparable for other ALL-F ensemble members. The LM evolution of global continental SAT, GST, ST$_{L8}$ and

ST$_{L15}$ anomalies relative to the 1850-2005 mean are shown. For ST$_{L15}$ non-filtered model output is represented and evidences the low pass filter influence of the heat conduction below the surface, whereas for SAT, GST and ST$_{L8}$ 31-yr running mean low pass filter outputs are shown. Subsurface temperature anomalies closely track SAT anomalies with relative small differences between them, thus indicating that air and soil temperature are coupled above multi-decadal time scales. The correlation coefficients for the 31-yr filtered series in Table 4 indicate high correlation (p<0.05) for the soil layers close to the surface that

diminishes slightly with depth due to the phase shift of the signal. Table 4 also indicates that the correlation considering the high-frequency variations (yearly; left column) is only high at the levels close to the surface, whereas at the deepest layer the correlation is low since the high-frequency variations are progressively filtered out and phase shifted as depth increases.

Despite the strong coupling between air and subsurface temperatures at global scales, the existence of a relative small offset between SAT and GST that grows backwards in time is evident for the annual (Fig. 1a) averages. This indicates a slight long-

term decoupling between SAT and GST. Previous works have argued that the nature of this offset arises from the changes in snow cover and its influence insulating the ground from very low air temperatures (e.g. Mann and Schmidt, 2003; Bartlett et al., 2004). González-Rouco et al. (2009) suggested that beside snow cover, soil moisture has also the potential of accounting for the thermal difference between SAT and GST. In order to explore the influence of such processes on the global SAT-GST relationship this analysis is extended considering winter and summer seasons (DJF and JJA hereafter) independently (Fig. 1c,e).

The offset between SAT and GST observed in the annual plot is apparent only in the DJF season. Indeed, the differences are larger during this season than in the annual data whereas the JJA evolution of SAT and GST anomalies is virtually identical. In addition, the correlation coefficients (Table 4) suggest slightly lower correlation for DJF than for JJA, even if SAT and GST are highly correlated and significant for both seasons, especially for the filtered values. The largest offset occurring in DJF





reinforces also the important role of snow cover in decoupling air and soil temperatures. Figure 1b,d,f further illustrates the strong relation between SAT-GST offset and the snow cover by displaying the 31-yr low pass filter outputs of the LM evolution of global snow cover and SAT-GST differences. Note that on the NH winter (DJF) and, due to its influence, also on annual averages, the decrease (increase) in snow cover lead to a decrease (increase) in the SAT-GST offset. This is due to the insulating

effect of snow that keeps GST close to zero while SAT can reach large negative values. Thus, an increase of snow cover leads to larger negative SAT-GST differences. For JJA on the contrary, an in-phase relationship is found at all time scales. Long-term trends change both in snow cover and in SAT-GST after the end of the 19th century. During the boreal summer increases of snow also enhance SAT-GST differences due to insulation of the ground from the warmer summer SAT; and conversely for snow cover decreases. This effect dominates the global average over that of the JJA austral winter during which SAT-GST

and snow cover changes experience an anti-phase relationship as described above. Therefore, the NH influences anti-phase covariability of snow cover and SAT-GST during DJF (detrended correlations, r=-0.52, p<0.05) and annual (r=-0.67, p<0.05) and in-phase covariability during JJA (r=0.62, p<0.05).

Dashed lines in Fig. 1b,d,f indicate the long-term trend in both SAT-GST offset and snow cover. Interestingly, the SAT-GST offset remains constant within the pre-industrial period but its long-term trend experienced a change during the industrial

period. The behavior is similar for the annual and seasonal cases with some difference in the timing. Such variations in the long-term relationship should be taken into account since they play an important role in decoupling SAT and GST at time scales relevant for the borehole theory. Nonetheless, this relative small change in the long-term trend has limited impact for the global SAT-GST coupling (Sect. 4.1).

The spatial variability of the relationship between SAT and GST gives further insights about the role of different processes

on the SAT-GST coupling. Figure 2 shows the spatial distribution of the differences between mean LM SAT and GST as well as the correlation coefficients for annual, DJF and JJA averages in the ALL-F ensemble. Maps have been obtained from the ALL-F$_2$ member; dots indicate that 80% of the ensemble members agree in delivering significant differences or correlation for a given grid-point. In the case of the annual temperatures (Fig. 2a), GST is generally warmer than SAT being the differences low in most of the globe (less than 2 °C) except in the NH mid- and high-latitudes areas where the differences are higher (up

to 15 °C). The correlation maps (Fig. 2b) provide a similar pattern with high and significant values in most the globe (>0.8 in regions located below 45° north) and lower correlations over NH mid- and high-latitudes; especially over the east of Siberia. A similar behavior is seen in the DJF season although more pronounced. During these months GST is much warmer than SAT reaching differences up to 30 °C at the northernmost part of North America and Eurasia (Fig. 2c). Differences are smaller at mid- and low-latitudes. The influence of the ocean over coastal areas providing larger SAT relative to GST is noticeable.

Likewise, the correlation is lower over northern snow covered areas (Fig.2d) while in the rest of the globe it remains high. On the contrary, Fig. 2e,f shows that during JJA, when the snow cover is scarce, the SAT-GST coupling is strong globally with temperature differences lower than 2 °C, and high correlation coefficients (>0.9). Consequently, the role of the snow cover in decoupling SAT and GST is highlighted. Positive correlation values are low over borderline areas where snow cover is more variable, thus producing variability in the SAT-GST offset and thereby altering the covariance structure. Close to these areas in

central Asia the high negative correlation within the Tibetan Plateau is noteworthy (see discussion below).





Figure 3 illustrates the SAT-GST decoupling due to the snow cover at the local scale for a particular grid-point as an example. The grid-point is located over a region with considerable snow cover during the boreal cold season (northeastern Russia). The DJF and JJA evolutions of SAT, GST, ST at different depths and the snow cover for the last 105 years from the ALL-$F_2$ simulation are shown. Note that in DJF the snow covers 100% of the grid-cell during almost the whole period. Thus the soil is

insulated and the difference between SAT and GST is ~-15 °C on average. The temperature of the deeper layers are presented in order to illustrate the amplitude attenuation and phase shift with depth of the temperature signal. Note that during DJF, GST is only slightly below 0 °C and the agreement of its variations with those of SAT is only noticeable in the largest changes of both while $ST_{L6}$ and deeper STs are above 0 °C. In JJA, SAT and GST are very similar and their low frequency variability propagates to deeper levels, all above 0 °C.

Some aspects of the SAT-GST spatial distribution deserves further attention. One of the most noteworthy is that over the Tibetan Plateau region temperature differences between SAT and GST are as large as in the mid and high NH latitudes for both annual and DJF (Fig. 2a, c). However SAT and GST are negative correlated (Fig.2b,d), being the only region of the globe where this occurs. For JJA the correlation nonetheless is positive and high (Fig. 2e). The nature of this opposite phase arises from discontinuous snow cover over this region during DJF. Usually the snow cover insulates the soil from the colder air, avoiding

the heat exchange with the atmosphere (as shown in Fig. 3). Nevertheless, discontinuous snow cover insulates the soil only partially, leading to this particular SAT-GST interaction. Figure 4 displays this behavior for a grid-point located over the Tibetan Plateau. SAT, GST, ST at different depths, snow cover and surface sensible heat flux (SHFLX) are shown. In DJF during periods of low snow cover the fraction of surface expose to the atmosphere allows for exchanging energy from the warmer soil to the colder air. Conversely, when snow cover is high, the large fraction of insulated soil reduces almost completely the heat

transfer from the soil to the atmosphere. Therefore with lower (higher) fractions of snow cover, higher (lower) heat transfer takes place with GST decreasing (increasing) and SAT increasing (decreasing). Indeed, there is a high negative correlation (-0.84, $p<0,05$) between the snow cover fraction and SHFLX which is the main way for energy dissipation within this region since latent heat fluxes (LHFLX) are negligible. Higher/lower albedo due to variations in snow cover fraction also contribute to the negative SAT-GST correlation over this region (not shown). During JJA on the other hand, snow cover is negligible thus

SAT-GST coupling is strong. Note that GST and all ST are above zero and GST is higher than SAT as it is warmed by radiative gain and transferring heat to the atmosphere, thus the positive correlation (0.5, $p<0,05$) between SAT-GST and SHFLX.

The spatial SAT-GST differences during JJA (Fig. 2e) depict other relevant aspects that have an influence on the SAT-GST relation at relative short-time scales. SAT is generally colder than GST globally however for JJA there are large areas inland, mainly located at the south-east of the United States, some parts of central and eastern Europe and eastern Asia with warmer

SAT relative to GST. Variations in LHFLX from DJF to JJA are driving this effect. Figure 5 shows that the areas where LHFLX increase in JJA relative to DJF are related to the same areas where SAT is higher than GST in JJA (Fig. 2e). Therefore there is a direct relation between the increase of evapotranspiration in JJA and the ground temperature response at these locations. The timeseries in Fig. 5 display this behavior for a grid-point located over southeastern of China. During JJA the surplus of energy due to higher solar radiation reaching the surface is mostly dissipated as latent heat leading to a net heat loss at the

ground surface. Note the anti-correlation between surface soil moisture and LHFLX (-0.46; $p<0,05$) during JJA as well as the



large SHFLX values that take place when soil moisture is lowest and evapotranspiration is limited. Therefore, the high rate of LHFLX contribute to cool the surface and GST tends to be lower relative to SAT. Soil water content also exhibits large changes during JJA in consistence with the large evapotranspiration (Fig. 5, top), thus providing the source of moisture that contributes to temperate SAT and cool GST. Large variations in evapotranspiration from DJF to JJA are also present at mid-latitudes of the

SH summer continents (America, Africa and Australia) although only a very limited impact on Fig. 2e is perceived; especially over the western coast. Over these regions high incoming energy impinging the surface during SH summer (not shown) supports high rate of latent heat fluxes, however as soil water becomes a limiting factor, more energy is dissipated as sensible heat and the ground surface is warmed. Therefore GST experiences higher temperature than SAT on average. Figure 5 also shows large evapotranspiration over the tropical rain forest in America and Africa, both in DJF and JJA that does not translate to posi-

tive SAT-GST differences. Over the rain forest, the energy fluxes at the surface do not vary significantly from DJF to JJA since incoming radiation is relatively constant throughout the year, and transferred to evaporation and evapotranspiration within the canopy while soils are well watered by precipitation to support the large amounts of evapotranspiration. This situation leads to a small range of variation both in SAT and GST with very small differences between them and higher GST relative to SAT.

Figure 2a,c,e shows similar higher SAT relative to GST also at some coastal areas both in JJA and in DJF. During DJF the effect

is mainly present in the NH mid-latitudes as, for instance, in most of the coasts of Europe, both the east and the west coasts of North America, as well as in Japan and the east coast of China. During JJA on the other hand, this behavior is seen mostly in the SH mid-latitudes, as in southern South America, South Africa and the south of Australia. Therefore there is a clear relationship between the winter season for each of the hemispheres and this SAT-GST temperature response in the CESM-LME simulations. Interestingly, higher SAT relative to GST is also evident at some coasts over tropical regions both in JJA and in

DJF mainly over southeastern Asia, the south of the Indian subcontinent, the Gulf of Guinea and some areas of South and Central America. In the CESM-LME the atmospheric grid-box of the coastal areas is partitioned into land and ocean fractions, For the areas with sea ice formation an additional sea ice fraction is considered (Neale et al., 2012). This configuration of the coastal grid-points leads to a partition of the energy fluxes at the surface into those of the land fraction and those of the ocean part. During the cold season such partition is determining the higher SAT warming relative to GST since the relatively low net

radiation that impinges the surface at mid- and high-latitudes limits the ground surface heating as well as the energy fluxes out of the land surface fraction. In contrast, the energy fluxes from the ocean surface to the air above are large mainly as a result of the temperature difference between the water and the air above, since the water is relatively warmer than the continental air in winter. Strong winds blowing predominantly onshore over most of the coast also contribute with advection of latent heat from the relatively warmer ocean. The dissipation of energy from the ocean fraction to the atmosphere warms the air so the net

effect is the higher SAT relative to GST in the winter season. Over the tropical coasts that exhibit the same behavior the energy fluxes out of the ocean fraction of each grid-point also contribute to the higher warming of the air relative to the ground surface. Nevertheless, although at these locations high rates of evapotranspiration all year long play also an important role since they generate evaporative cooling of the ground surface as in the example describe in Fig. 5.

The different examples used to illustrate the most important processes that may influence the air and soil temperature relation-

ship at short time scales depict also relevant information about the propagation with depth of the annual cycle. For instance, at





the grid-point located over the south east of China in Fig. 5, both in DJF and JJA, the temperature offset between contiguous levels is noticeable with a gradient of about 5 °C in the first meter of the ground and of about 10 °C down to the lowest level. Comparable pictures with some differences in the magnitude of gradients can be seen in the previous figures.

Therefore, it is interesting to understand also the propagation below the surface. GCMs simulate purely conductive regimes,

and the temperature variations that propagate to deeper soil layers are established at or near the ground surface (Smerdon et al., 2003). Thus it is important to asses the propagation of the temperature signal within the shallow subsurface. This issue is addressed by analyzing the relationship between GST and $ST_{L8}$.

Figure 6 (left) provides an spatial view of the temperature differences between GST and $ST_{L8}$ for annual, DJF and JJA. The correlation is also shown in the right panels. SAT-GST differences, for DJF and JJA show the yearly cycle of temperature

with negative (positive) SAT-GST for the NH (SH) in DJF and viceversa for JJA, illustrating the conductive regime within the shallow subsurface. The annual temperature differences are low and the correlation is high almost globally as it is the balance between the respective patterns in summer and winter. However, the northernmost part of the globe exhibits larger temperature differences (between 4 and 5°C) and lower correlation coefficients. The DJF and JJA patterns show that the annual offset and correlations for this part of the globe are mostly the result of the larger weight of those in JJA given that during this months

the temperature differences for a latitudinal band at ca. 60-70 N° are as large as 15°C and the correlation coefficients are close to 0. García-García et al. (2016) describe a similar behavior in some of the GCMs used in their analysis, detected over areas where frozen ground persists during JJA. Indeed, the nature of the large departure in the temperature response at the shallow subsurface at these locations arises from non-conductive processes related to latent heat release/uptake of freezing and thawing of the water content above 1 m depth that may account for the subsurface heat transfer (Kane. et al., 2001).

To illustrate this mechanism, Fig. 7 shows the temperature evolution of SAT, GST and ST at different depths as well as the soil ice content (SIC) in the upper soil layers for a grid-point located at the north of Canada. Over these areas the SIC in the upper 1 m depth increases (decrease) during the cold (warm) season. During JJA SAT and GST increase/decrease at the same rate since no ice is present at the ground surface so it is warmed by radiative gain and heat is transferred to the atmosphere. However, for deeper soil layers the energy available is employed in melting the SIC (note the lower SIC in L6 during JJA relative to DJF) and

latent heat is required so these layers do not experience a temperature increase as the shallowest ones. Therefore the temperature at L8 (~1 m depth) is kept below/near 0°C during the warm season due to the the zero-curtain effect (Outcalt et al., 1990), while GST is centered around 12°C, leading to differences of ca. 15°C between GST and $ST_{L8}$ and low correlation coefficient (0.28 for this grid-point) during JJA. In turn, during DJF SAT distributes around -35°C and the frozen ground experiences skin temperatures of about -8°C and of about -4°C at 1 m depth. Consequently, the temperature offset between GST and $ST_{L8}$ is

largest in summer. According to this, there are some non-conductive processes associated with permanent frozen soils in the shallow subsurface that are included in the CLM4 parametrization (Lawrence et al., 2011) and that play an important role on the heat transport. The higher SIC in L8 during JJA relative to DJF (Fig. 7) arises from the fact that freezing of the active layer begins in late autumn and, due to the release of the latent heat of fusion, the freezing front is inhibited (Hinkel et al., 2001) and only reach L8 in spring when SIC at this layer peaks. Then, as the thaw front penetrates downward in summer, ice

in the shallowest soil is melted, but it does not reach L8 until autumn, when SIC at this layer is lowest. Therefore due to the





thawing/freezing processes, seasonal changes at the upper and deeper subsurface levels are phase-shifted.

## 4.1 SAT-GST long-term changes

The mechanisms that have been described have an impact on the coupling between SAT and GST at short-time scales but
they do not affect the long-term SAT-GST association if they are stationary since its influence would be constant at long time
scales. However if such mechanisms experience variations with time, the SAT-GST relationship would also change with time,
thus the thermal signature imprinted in the subsurface would not be representative of the long-term SAT variations (Bartlett
et al., 2005). Figure 1b illustrates the existence of a constant offset between SAT and GST within the preindustrial period that
changes during the industrial times indicating a variation in the long-term SAT-GST relationship. This may be relevant in the
interpretation of borehole climate reconstructions since it may induce a long-term decoupling between SAT and GST in the
CESM-LME. At global scale the change in the long-term SAT-GST offset have an impact of about 0.05°C (Fig. 1a,b) and thus,
do not seem to be, at these scales, very relevant. However, the impacts could be larger for other GCMs with higher climate
sensitivity or a different representation of surface processes that may contribute to decouple GST from SAT (e.g. snow cover).
Likewise, within the CESM-LME simulations impacts on decoupling may be important at regional or local scales.

To examine the spatial distribution of the long-term SAT and GST evolution during industrial times, we evaluate the linear
trends of both of them independently during this period at every land model grid-point. Besides the ALL-F ensemble we have
also considered in this analysis the anthropogenic single-forcing ensembles (Fig. 8), bearing in mind their potential influence
on the processes that modulate the relationship between SAT and GST, as for instance: variations in snow cover, soil moisture
and albedo, among others. The results in this section are shown considering information from all members of the ensembles
(see details in Table 3). For specific examples one of the members will be used as indicated accordingly in figure captions. Fig-
ure 8a,b describes a predominant warming for both SAT and GST in the ALL-F ensemble with the largest values distributing
over north-west North America, north and central Eurasia, north east Africa and southern South America. Interestingly, there
are also regions showing negative trends like southeastern China, the north of the Black and Caspian sea regions, over Pakistan,
some relative small areas of central and south of Africa and over Brazil. The warming trend pattern can be to a large extent
explained if the 1850-2005 trends are calculated on the basis of the GHG-only ensemble (Fig. 8c,d) which is consistent with
the global warming pattern due to the influence of GHGs (Hartmann et al., 2013). Indeed, if only the contribution of GHGs
is considered the warming would be higher and globally distributed. Figure 8 also indicates that the cooling in the ALL-F
ensemble is mainly driven by the contribution of the LULC and OZ/AER external forcings. For instance, the cooling trends
over the Baltic sea and the north of the Black and Caspian seas that dominate the SAT and GST cooling trends during the
industrial period in the ALL-F ensemble are the result of the influence of LULC changes (Fig. 8e,f) with additional contribu-
tions of OZ/AER (Fig. 8g,h). In addition, the negative trends of both SAT and GST over some areas of Africa as well as over
the north east of Brazil are also detectable in the LULC-only ensemble (Fig. 8e,f). Likewise, the OZ/AER-only ensemble also
contributes to the cooling over Brazil, and the strong negative trends observed at south-east China in the ALL-F ensemble are
clearly identifiable in this ensemble (Fig. 8g,h).





Although the general pattern of cooling/warming during industrial times is broadly similar for SAT and GST, it differs substantially in some regions. Note that there are considerable differences in the amplitude of warming trends in SAT and GST over Fennoscandia as well as at the northernmost part of North America in the ALL-F ensemble (Fig. 8a,b). Similarly, over some areas of central and eastern Europe SAT and GST industrial trends have different sign. There are also considerable differences

in the amplitude of the cooling in SAT and GST over northeastern Brazil. Such dissimilar behaviors of SAT and GST during the industrial period are connected to variations in the energy fluxes at the surface in response to changes in the land surface characteristics due to the influence of the external forcings during this period.

For a more detailed analysis of SAT and GST long-term relationship, we applied a two-phase regression model (see Sect. 3) during the complete LM at every land model grid-point to the differences between SAT and GST in the ALL-F ensemble as

well as in the GHG-only and LULC-only ensembles (Fig. 9). In the case of the OZ/AER-only ensemble the linear trends of the differences between SAT and GST during the industrial period are presented in Fig. 10 since this set of simulations span solely from 1850 to 2005 CE. Figure 9a,b,c shows the year-of-change of the long-term SAT-GST trends for the ALL-F, GHG-only and LULC-only ensembles respectively. Figure 9d,e,f (9g,h,i) represents the magnitude of the change before (after) it takes place. Note that except for some areas, most continental regions experience statistically significant dates of change in phase

with the start of the industrial period, during the 19th century (darker blue). For all of them SAT-GST offset trends before the change are negligible or slightly negative in the ALL-F ensemble (Fig. 9d), indicating a stable relationship between SAT and GST globally before industrial times. This characteristic is also evident in the GHG and LULC-only ensembles (Fig. 9e,f). The strong positive or negative trends before the change (Fig. 9d,e,f) correspond to early dates of change around the 10th and 11th centuries (light green colors in the year-of-change maps) as a result of a change in the simulated differences between SAT and

GST early in the LM and a stable behavior in GST-SAT differences thereafter. Note that for all of these regions the trend after the change is negligible and none of them is statistically significant (Fig. 9c,f,i).

The trends after the change on the other hand, indicate significant regional variations in the long-term SAT-GST relationship. In general annual SAT minus GST yields negative values (as shown in Fig. 2a) with the exception of the coastal areas as explained in Sect. 4. Thus, for continental areas (SAT-GST<0), positive (negative) trends indicate that differences tend to get

smaller (larger) in absolute values, and conversely for the limited coastal areas where SAT-GST differences are positive. Figure 1b allows for visualization of this behavior. Note the positive trend after the change when the difference between SAT and GST anomalies becomes smaller with time. Regionally, several circumstances account for impacting SAT-GST long-term coupling. On the one hand, decreasing SAT-GST differences over land may emerge from two conditions. Firstly, when there is a higher warming rate of SAT relative to GST as depicted in Fig. 8a,b over the northernmost part of North America, Fennoscandia,

northeast Russia and some areas of central Eurasia. Secondly, when there is a cooling of both SAT and GST but the later decreases at a higher pace as described in Fig. 8a,b for the region of northeastern Brazil and some areas of Africa. Note that these two scenarios are represented in Fig. 9g with positive trends over these regions after the change. On the other hand, the increase in SAT-GST difference arises either from the effect of rising GST in the presence of stable/decreasing SAT or due to the higher warming rate of GST relative to SAT. The former case is appreciated in Fig. 8a,b for the areas of central and eastern

Europe as well as eastern U.S whereas the latter is found over the Indian subcontinent and southeastern Asia. Note that both





cases are represented in Fig. 9g with negative trends.

Trends of differences between SAT and GST from the GHG-only and the LULC-only ensembles help understanding the relative contributions to the long-term variations seen in the ALL-F simulations. For instance, the GHG-only ensemble (Fig. 9h) shows similar positive trends as Fig. 9g over northern North America, Fennoscandia, northeast Russia and central Eurasia,

though with a much larger magnitude and geographical extension. Correspondingly, negative trends after the change in the LULC-only ensemble (Fig. 9i) are comparable to those in Fig. 9g for the regions of central and eastern Europe, eastern U.S., the Indian subcontinent and southeastern Asia. Additionally, the positive values over Brazil as well as over central and southern Africa in Fig. 9g are also depicted in Fig. 9i.

Interestingly, the two-phase regression analysis does not expose any variation in SAT-GST long-term relationship over south-

eastern China, where the linear trends during the industrial period show a relative strong decrease in both SAT and GST in the ALL-F and the OZ/AER-only ensembles (Fig. 8a,b,g,h). Furthermore, the linear trend of SAT-GST differences during industrial times for the OZ/AER-only simulations (Fig. 10) does not exhibit any SAT-GST decoupling over this region either. This suggests that the dominant effect of OZ/AER forcing on the SAT and GST responses over this region is not affecting their long-term coupling. Nonetheless, Fig. 10 illustrates some interesting aspects of the SAT-GST relationship in the OZ/AER-only

ensemble such as the negative contribution to the SAT-GST trends over North America, northern Europe, the Tibetan Plateau and central Asia. Additionally, the positive trends over northern Siberia are also notable as well as the positive values over some relative small areas of central and eastern Africa, the coast of Angola and eastern Brazil. Although the bulk of these SAT-GST responses depicted in Fig. 10 does not translate to SAT-GST long-term decoupling in the ALL-F ensemble, they play an important role either in counteracting the influence of other external forcings or contributing to decoupling-related

processes over some regions.

The following paragraphs aim at providing an insight into the relative contribution of the individual forcings and the associated physical mechanisms to the variations of the long-term SAT-GST association detected in Figs. 9 and 10.

In the cases of the long-term variations due to the LULC influence, changes in vegetation cover can be considered the main driving factor since they alter the radiative fluxes and water cycling at the surface due to the modification of the physical prop-

erties such as albedo, roughness and evapotranspiration (Pongratz et al., 2010). Figure 11 gives an example of how long-term changes in the energy fluxes at the surface due to LULC changes do impact the SAT-GST coupling at long time scales. It shows the 31-yr low pass filter outputs of SAT, GST, reflected shortwave radiation (RSW) and SHFLX evolution for a characteristic grid-point over the Great Lakes region (U.S.) where a warming of GST relative to SAT is simulated in CESM-LME during the industrial period. Results are shown for one of the members of the ALL-F, LULC-only, GHG-only and OZ/AER-only ensem-

bles. Around 1800 CE SAT tends to decrease whereas GST tends to increase in both the ALL-F and LULC-only simulations, while GHG-only and OZ/AER-only simulations do not display the same behavior that produces larger differences between SAT and GST represented by negative trends in Fig. 9d. At the same time RSW and SHFLX exhibit large long-term variations in the ALL-F and LULC-only simulations. Therefore, this modification of the long-term SAT-GST relationship is clearly the response to LULC changes. Such variations in the surface energy fluxes over this region are likely a response of vegetation

replacement from forested areas to grassland or croplands. Forested landscapes dissipate SHFLX more efficiently to the atmo-





sphere due to a higher surface roughness than open fields (Jackson et al., 2008). In addition, lower vegetation types have higher reflectivity than forests. All the previous contribute to SAT decreases over these regions, especially in winter. Furthermore, deforestation process at mid- and high-latitudes tends to positively feedback with increases in snow cover (Anderson et al., 2011). This type of changes in LULC contribute to increasing albedo which is reinforced by changes in snow cover at these

latitudes. Additionally, higher DJF snow cover tends to increase insulation of the soil from the cold overlying air. Therefore, the different temperature response of SAT and GST is related to higher albedo that leads to cooling the atmosphere, the reduction of SHFLX from the ground surface to the atmosphere and the additional snow cover positive feedback. This particular LULC process is important for corrupting the SAT-GST coupling at time scales relevant for the borehole theory (centennial) since the thermal signature recorded in GST during the industrial period would not be representative of the past long-term SAT variations

in regions where this effect is dominant. For the areas of central and eastern Europe where a GST warming relative to SAT is also observed in Fig. 9c the mechanisms are similar to those described in Fig. 11 because these areas were also subject to intense transformation from forested areas to cropland at the beginning of the industrial period according to the LULC forcings considered in the CESM-LME (Pongratz et al., 2008; Hurtt et al., 2009).

Changes in vegetation cover are also important for long-term SAT and GST temperature differential response over tropical

regions although the driver mechanisms are different than those at mid- and high-latitudes (Lee et al., 2011), and they deserve being considered. At the northeast of Brazil both SAT and GST have negative trends in both the ALL-F and the LULC-only simulations during the industrial period (Fig. 8a,b,e,f). However, the decrease of GST is much larger than the one of SAT as represented in Fig. 9c,i with positive trends after the change.

Figure 12 shows the temporal evolution of SAT, GST, RSW and LHFLX for a grid-point located at the northeast of Brazil. At

the end of the 18th century GST drops sharply whereas SAT slightly decreases. Likewise, RSW and LHFLX experience significant changes at the same time as a result of the modification in the surface characteristics. Such changes are present solely in the All-F and LULC-only simulations whereas GHG-only and OZ/AER-only ones show no differences in their evolution. The changes observed at this location in the energy fluxes likely correspond to transitions from open lands to a forested area (reforestation or afforestation) leading to lower albedo and higher evapotranspiration rate as is shown in Fig. 12. This situation

leads to an apparent long-term cooling of GST relative to SAT at this location. The temperature response over this area is influenced by different mechanisms. First, the conversion from lower- to higher-type vegetation reduces the solar radiation that impinges on the surface and GST decreases by a radiative effect. Second, forested lands usually have lower albedo thus absorb more shortwave radiation (Zhao and Jackson, 2014). This surplus of energy is balanced by the increase in transpiration, and consequently GST decreases also by a non-radiative process. The latter is especially important in humid climates (von Randow

et al., 2004) as the one in this example. According to this, there is a net heat loss at the ground surface with a higher decrease in GST relative to the overlying air. The SAT-GST coupling becomes strong again after some period when the new vegetation cover reaches a stable estate by mid 20th century. For the areas of Africa where there is also a cooling of GST relative to SAT (Fig. 9g,i), the mechanisms are comparable to those describe for Fig. 12.

Over some of these tropical regions there is also a contribution from the OZ/AER forcing to the SAT-GST response. Note

that Fig. 10 shows positive trends over Uganda, the coast of Angola and over the east of Brazil, also noticeable in Fig. 9c.



At these regions the incoming solar radiation is reduced due to the effect of aerosols as they are an important element for cloud formation, thus contributing to a higher reflectivity of solar radiation (Tao et al., 2012). The description of the specific processes related to aerosols-cloud interaction goes beyond the scope of this study, therefore only the influence on the energy balance at the surface is addressed. As lower shortwave radiation impinges the surface, the energy gain decreases and therefore

the ground surface heating is lower. The reduction in the energy gain at the surface is compensated by a lower dissipation via sensible heat whereas the fluxes of latent heat remains relatively constant or even increases in some areas due to a higher moisture as a result of increased precipitation. To illustrate this mechanisms Fig. 13 shows the temporal evolution of SAT, GST, SHFLX, LHFLX and surface incoming shortwave radiation (SSW) for a grid-point located at the region of Uganda. Note in the ALL-F simulation the reduction in SSW after 1900 CE that leads to a decrease in SHFLX. Interestingly, for this location

LHFLX experiences an increase at the same time as a result of increase precipitation (not shown) thus providing the source of moisture for evapotranspiration. This situation leads to a higher decrease in GST relative to SAT due to a net loss of energy at the ground surface. Similar SAT and GST responses are also observed in the OZ/AER-only and LULC-only simulations, whereas in the GHG-only, SAT and GST increase.

The SAT-GST decoupling processes described above for individual grid-points are important also at larger spatial scales. Figure

14(left) shows an extension of the mechanism depicted in Fig. 12 including a larger area over the northeast of Brazil (between 1°S-11°S and 47°W-35°W). The negative trend since the 18th century is less accentuated for SAT than for GST (-0,014 and -0,053 °C/decade respectively in the All-F simulation and -0.007 and -0.033 in the LULC-only one) indicating strong contribution of past LULC changes. The same air-subsurface temperature response occurs in other tropical and subtropical areas such as the east of Africa.

The regional analysis is extended to the area of southeastern China in order to illustrate additional information about the influence of different external forcings on the SAT-GST relationship. As discussed before, the negative trends for both SAT and GST over this region during the industrial period represented in the ALL-F and OZ/AER-only ensembles (Fig. 8a,b,g,h) do not entail a corruption of the SAT-GST long-term coupling. Figure 14(right) allows for an insight into this particular effect and the role of the OZ/AER forcing on the air and soil temperature responses over this region during industrial times. The evolution

of the 31-yr filtered outputs of SAT, GST, RSW, LHFLX and SHFLX are presented. Note that the negative long-term trend within the industrial period is seen only in the ALL-F and the OZ/AER-only simulations. Similarly, in both of them there is a reduction in the RSW as well as in both the LHFLX and SHFLX. On the other hand, the SAT and GST evolution at industrial times in the GHG-only and LULC-only simulations does not follow the same path depicted in the ALL-F, thus highlighting the dominant influence of OZ/AER forcing. In this case the variations of the energy fluxes at the surface depend on the reduction

of the incoming shortwave radiation as a response of the anthropogenic aerosol-cloud interaction rather than by modifications of the land surface properties. Therefore the decrease in the energy that impinges the surface is balanced by a decrease of both the sensible and latent heat fluxes. Hence, the air-soil interactions are not significantly altered and the SAT-GST relationship remains stable.

Although the majority of the important long-term variations in the SAT-GST relationship at regional and local scales observed

in the ALL-F ensemble (Fig. 9c) are induced by LULC changes, there are some regions in which the GHG forcing is the main



driver for long-term SAT-GST decoupling. The positive trends after the change over Fennoscandia, northeast Russia and north of North America observed in the ALL-F ensemble (Fig. 9c) can be to a large extent explained by the influence of GHG-only ensemble inasmuch as a broadly similar picture is also portrayed over these regions in Fig. 9f. In the case of the GHG-only ensemble, the strong warming of SAT relative to GST over these regions is driven by the increasing air temperature during

industrial times due to the the positive radiative forcing of GHGs in the presence of a considerable long-term reduction in simulated snow cover. González-Rouco et al. (2009) showed that such an scenario would lead to a higher exposure of soil to cold winter air, therefore the soil would partially record colder temperatures, previously prevented by the snow cover insulating effect. In the ALL-F ensemble this effect is damped for the fact of considering additional forcings that keep the snow cover relatively constant during industrial times. For instance, the contribution of the OZ/AER forcing is particularly important for

counteracting the effect of GHGs since it leads to colder climate conditions due to its negative radiative forcing. Note the strong negative trends in Fig. 10 over North America, northern Europe and the Tibetan Plateau that partially balance the effect of the GHGs over these regions. Nonetheless, in the ALL-F there is still an overall SAT warming relative to GST since the relatively stable snow cover is insulating the soil from a warmer SAT reducing the overall offset between them (Bartlett et al., 2005).

Figure 15 gives further insights into the interactions between anthropogenic forcings, their influence on global snow cover and

consequently on the long-term SAT-GST relationship in the CESM-LME. The LM evolution of global SAT minus GST (top) and the annual global snow cover (bottom) for the ALL-F, LULC-only, GHG-only and OZ/AER-only simulations are shown for one member of each ensemble. Similar results are obtained if other members are selected. On the one hand, when the GHG-only ensemble is considered the global SAT-GST offset experiences a sharp long-term decrease in absolute value at the start of the industrial period as well as a strong long-term reduction in global snow cover. In fact, the correlation between changes in

the SAT-GST offset and snow cover in the GHG-only ensemble member is high -0.93 (p<0,05). On the other hand, the overall effect of the LULC-only ensemble is a relative small increase in the global snow cover mainly due to deforestation at mid and high latitudes as well as due to the negative radiative forcing of LULC as Earth's albedo increases (Myhre et al., 2013), leading to a small increase in the global SAT-GST offset. In the same way the OZ/AER-only ensemble shows an increase in global snow cover while the SAT-GST offset in industrial times exhibits a relatively slight increase. The interaction between

different external forcings in the ALL-F ensemble leads to an stable snow cover during the industrial period since the sharp decrease in snow induced by the GHG forcing is partially compensated by the counteracting effect of the LULC and OZ/AER forcings. Additional forcings such as volcanic may also contribute to counteracting GHGs effect at multi-decadal timescales (not shown). Consequently, in the presence of a warmer climate there is a difference in the warming rate of SAT and GST in industrial times at global scale in the ALL-F ensemble member (0,025 and 0,018 °C/decade respectively). This scenario

leads to a net effect of long-term decrease in the SAT-GST differences starting around 1800 CE as discussed in Fig. 1 and also evident in Fig. 15.





## 5   Conclusions

This work evaluates the stationarity of the coupling between SAT and GST temperatures as simulated by the CESM in an ensemble of experiments spanning the LM. The initial motivation for this work roots on previous literature (González-Rouco et al., 2006, 2003, 2009; García-García et al., 2016) that addresses the realism of the borehole hypothesis for climate recon-

struction, namely, that SAT and GST vary synchronously and that reconstructing past GST changes from borehole temperature profiles is a good proxy for past SAT variations. The use of the CESM-LME allows for analyzing the influence of forcing changes on the SAT-GST covariability, both individually and as a group, by considering the different all-forcing and single-forcing ensembles. Additionally, having several experiment ensemble members for each given forcing type, allows for disentangling the effects of internal variability from that of the forcing response. Ultimately, the coupling between SAT and GST is

assessed at global and also at regional/local scales. This last purpose needs considering different mechanisms that contribute to SAT-GST variability within different climate types. In doing so, a variety of factors and conditions that contribute to the surface energy balance with different mechanisms is provided.

The CESM-LME shows that at global scale the SAT-GST coupling is strong above multi-decadal timescales since GST tracks SAT throughout the LM, as found in previous works. However in spite of the strong coupling, the CESM-LME also reflects that

the SAT-GST relationship has not remained constant along the whole LM at these spatio-temporal scales. Hence, the nature of such variation is evaluated.

Globally, snow cover is the most important agent in modulating the connection between SAT and GST. Therefore the variation of the SAT-GST relationship described by the CESM-LME simulations should be in principle driven by variations of global snow cover. Nevertheless, the simulated snow cover remains relatively stable at the time when SAT-GST coupling varies, thus

this change can not be solely explained by the influence of the snow cover. With this in mind, we explored with some detail different processes that may influence the SAT-GST relationship at different spatio-temporal scales. Firstly, we address processes acting at seasonal time scales that were identified from an spatial analysis of the SAT-GST differences and correlations. Secondly, the long-term evolution of the SAT-GST relationship is evaluated in the ALL-F, LULC-only, GHG-only and OZ/AER-only ensembles.

Several processes over different regions either relevant during winter/summer play an important role on impacting the SAT and GST coupling. Some examples are: snow cover over mid and high latitudes, discontinuous snow cover over the Tibetan Plateau region and seasonal variations in the energy fluxes at the surface. Although these processes are important for disrupting the SAT-GST relationship at seasonal scales, they have no implications on the long-term coupling if they are stationary. Nonetheless, if they experience variations with time the SAT-GST long-term relation may be impacted.

As discussed in Sect. 4.1 some of the anthropogenic external forcings have the potential to impose long-term variations on processes that regulate the relationship between SAT and GST. Among them, the LULC changes are the most important ones since they modify the energy fluxes at the ground-air interface, and consequently corrupt the SAT-GST coupling locally and regionally at various time scales. An example is the response to the deforestation processes triggered by the expansion of agriculture mainly during the industrial period at mid and high latitudes, where SAT and GST long-term coupling is impacted





due to the variations in the albedo, surface roughness and hydrology. Similar decoupling processes related to LULC changes
are found over different regions around the globe, as the one described in Sect. 4.1 over northeastern Brazil, some areas over
Africa and over the Indian subcontinent. All of them driven by the long-term modifications of the energy fluxes at the surface
either from increased evapotranspiration, reduced energy dissipation via sensible heat and others. Besides these kind of de-
coupling processes induced by individual forcings, the interactions of a variety of mechanisms and feedbacks from different
external forcings can also exert an influence on the long-term SAT-GST relationship at different spatial scales. For instance,
the effect of GHGs leads to a reduction of the snow cover during industrial times that is counterbalanced by the opposite effect
of both LULC and OZ/AER forcings. As a consequence, the snow cover remains relatively stable over some regions during
the industrial period in the presence of a warmer climate. This situation leads to a difference in the SAT and GST long-term
evolution during the industrial period, since the snow cover is insulating the soil from a warmer SAT. This effect is present
over the NH high-latitudes of North America, Fennoscandia and northeastern Russia. Indeed, at global scale the combination
of a steady snow cover under warmer climate conditions is the dominant effect for explaining the variations in the long-term
SAT-GST relationship.

Our findings indicate that the assumption of a strong relation between SAT and GST may be impacted from local to regional
scales by different mechanisms especially by the influence of LULC changes due to the modification of the energy balance at
the surface. Therefore, the interpretation of temperature reconstructions from borehole measurements at this spatial scales must
consider LULC changes as a source of possible bias. The effects of additional external forcings may also exert some influence
on processes such as variations in the snow cover, hydrology and other land surface properties, that may in turn feedback on
the SAT-GST long-term coupling. At global scale the influence of such local and regional decoupling processes is only ca.
0.05°C, hence the SAT-GST coupling at this spatial scale is supported by the CESM-LME.

*Competing interests.*   The authors declare that they have no conflict of interest.

*Acknowledgements.*   This research was supported by an FPI grant: BES-2015-075019, from the the Spanish Ministry of Economy, Industry
and Competitiveness. We gratefully acknowledge the IlModels project, CGL2014-59644-R. We also thank Bette L. Otto-Bliesner for some
personal communications regarding several aspects of the methodological design of the CESM-LME.



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



**Table 1.** Soil layers and node depths in the CLM4. Note that the node depth, that are the depth at which the thermal properties are defined for soil layers (Oleson et al., 2010), do not necessarily coincide with the center of the layer depth.

| Layer | Layer depth (m) | Node depth (m) |
|-------|-----------------|----------------|
| L1    | 0.017           | 0.007          |
| L2    | 0.045           | 0.027          |
| L3    | 0.090           | 0.062          |
| L4    | 0.165           | 0.118          |
| L5    | 0.289           | 0.212          |
| L6    | 0.492           | 0.366          |
| L7    | 0.828           | 0.619          |
| L8    | 1.382           | 1.038          |
| L9    | 2.296           | 1.727          |
| L10   | 3.801           | 2.864          |
| L11   | 6.284           | 4.739          |
| L12   | 10.377          | 7.829          |
| L13   | 17.125          | 12.925         |
| L14   | 28.252          | 21.326         |
| L15   | 42.103          | 35.177         |



**Table 2.** Simulations and LM external forcing reconstructions used in CESM-LME. The single forcing simulations cover from 850 to 2005 CE except those of anthropogenic ozone and aerosols that span the period 1850 to 2005 CE. Legend for external forcing: SOL, changes in total solar irradiance; VOLC, volcanic activity; GHG, concentrations of the well-mixed greenhouse gases $CO_2$, $CH_2$, and $N_2O$; LULC, land use land cover changes; (ORB) orbital variations; and OZ/AER, anthropogenic ozone and aerosols

| Forcing | No. of simulations | Reference |
|---|---|---|
| ALL-F | 10 | - |
| SOL | 4 | Vieira et al. (2011) |
| VOLC | 5 | Gao et al. (2008) |
| GHG | 3 | MacFarling Meure et al. (2006) |
| LULC | 3 | Pongratz et al. (2008) dataset, spliced to Hurtt et al. (2009) at 1500 CE. The only plant functional types (PFTs) that are changed are those for crops and pasture; all other PFTs remain at their 1850 control prescriptions |
| ORB | 3 | The CESM model adjusts yearly orbital position (eccentricity, obliquity and precession) following Berger et al. (1993) |
| OZ/AER | 2 | Fixed at the 1850 control simulation values until 1850 and then include the evolving anthropogenic changes to 2005. Stratospheric aerosols are prescribed in the model as a fixed single-size distribution in the three layers in the lower stratosphere above the tropopause. The ozone forcing is from the Whole Atmosphere Community Climate Model (WACCM) |

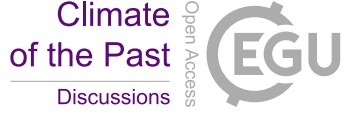



**Table 3.** Simulations used in this study from the CESM-LME. The first and second columns present the Acronyms used in this manuscript for ensembles and ensembles members respectively. The id of the original experiment files is provided in column 3.

| Ensemble acronym | Ensemble member | Simulation id |
|---|---|---|
| ALL-F | ALL-F$_i$ | b.e11.BLMTRC5CN.f19_g16.0i |
| | i=1,...,10 | i=01,...,10 |
| GHG-only | GHG$_i$ | b.e11.BLMTRC5CN.f19_g16.GHG.00i |
| | i=1,2,3 | i=1,2,3 |
| LULC-only | LULC$_i$ | b.e11.BLMTRC5CN.f19_g16.LULC_HurttPogratz.00i |
| | i=1,2,3 | i=1,2,3 |
| OZ/AER-only | OZ/AER$_i$ | b.e11.BLMTRC5CN.f19_g16.OZONE_AER.00i |
| | i=1,2 | i=1,2 |





**Table 4.** Temporal correlation coefficients between last millennium SAT-GST, SAT-ST$_{L8}$ and SAT-ST$_{L15}$ anomalies relative to the 1850-2005 mean for the experiment shown in Fig. 1. The left side indicates the correlation at yearly resolution whereas the right side shows 31-yr low-pass filter outputs of SAT, GST, ST$_{L8}$ and ST$_{L15}$ for annual, DJF and JJA. Coefficients highlighted in bold are statistically significant at p<0.05

| | Yearly | | | 31-yr filtered | | |
|---|---|---|---|---|---|---|
| | Annual | DJF | JJA | Annual | DJF | JJA |
| **SAT-GST** | **0.96** | **0.85** | **0.99** | **0.98** | **0.94** | **0.99** |
| **SAT-ST$_{L8}$** | **0.95** | **0.81** | **0.94** | **0.98** | **0.93** | **0.96** |
| **SAT-ST$_{L15}$** | 0.28 | 0.27 | 0.14 | **0.86** | **0.85** | **0.65** |





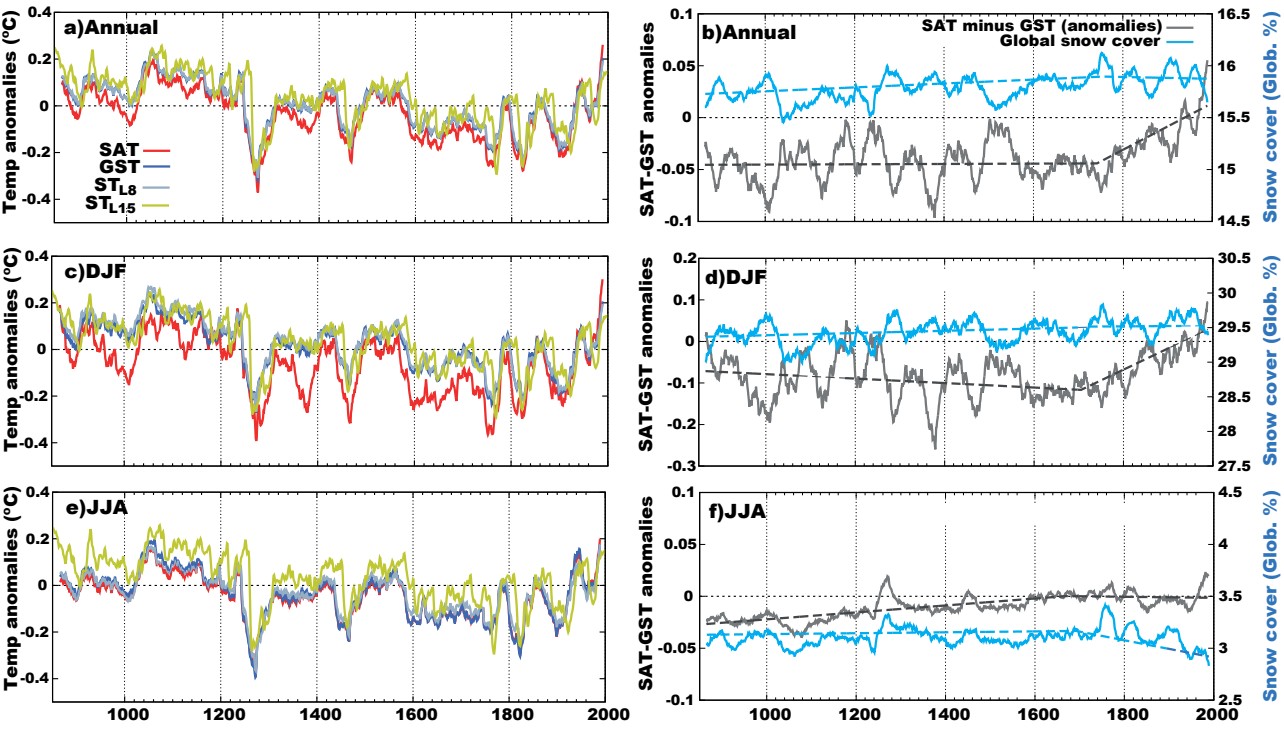

**Figure 1.** Left. (a) Annual, (c) DJF and (e) JJA global LM evolution of SAT, GST, $ST_{L8}$ and $ST_{L15}$ anomalies relative to 1850-2005 CE for the ALL-$F_2$ ensemble member (Table 3; Otto-Bliesner et al., 2016) . Right. (b) Annual, (d) DJF and (f) JJA global LM evolution of SAT-GST offset (SAT minus GST anomalies) and global percentage of snow cover. Dashed lines are the result of a two-phase regression, indicating the linear trend that represents the best fit to the data before and after the estimated point of change. For snow cover, dashed lines represent linear fits to the data using the change points found for SAT-GST. All series are 31-yr moving average filter outputs except for $ST_{L15}$ which is represented as the direct model output.





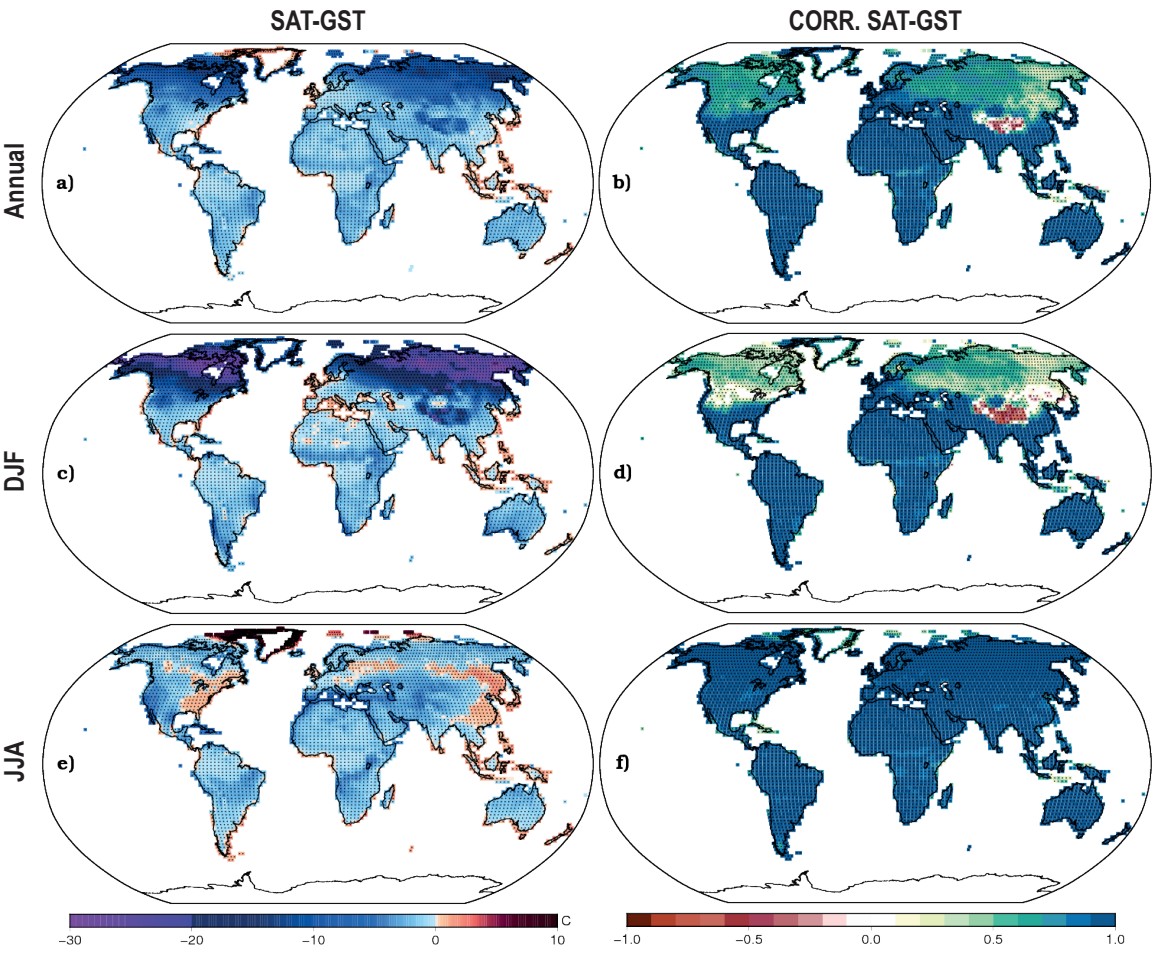

**Figure 2.** Left. (a) Annual, (c) DJF, (e) JJA spatial distribution of SAT minus GST differences during the 850-2005 CE period. Right. (b) Annual, (d) DJF, (f) JJA spatial distribution of the correlation coefficients between SAT and GST for the same period. Black dots indicate that 80% of the members within the ALL-F ensemble deliver statistical significance (p<0.05).





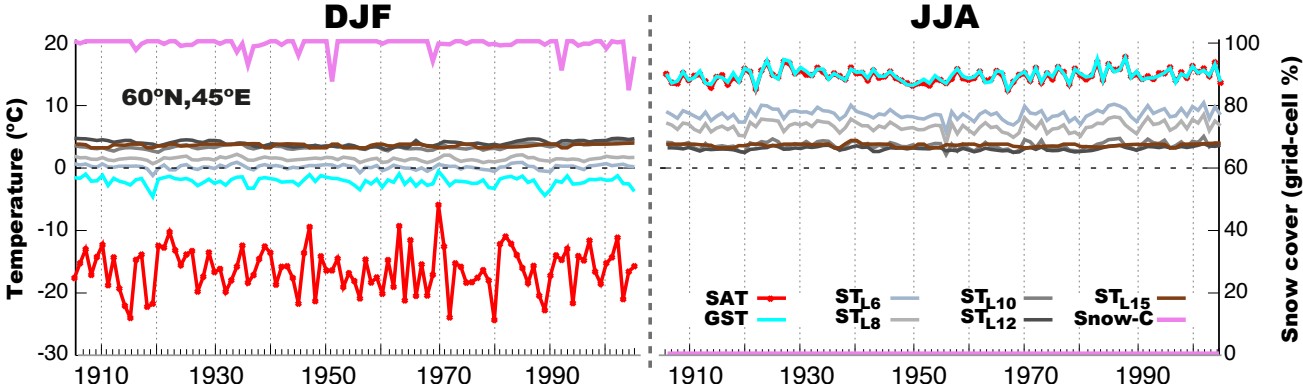

**Figure 3.** Evolution of DJF and JJA SAT, GST, $ST_{L6}$, $ST_{L8}$, $ST_{L10}$, $ST_{L12}$ and $ST_{L15}$ and percentage of snow cover during the period 1900-2005 CE for a grid-point (north-east of Russia; 60 °N, 45 °E) from the ALL-$F_2$ simulation (Table 3) where snow cover is characteristic during the cold season. Dashed lines indicate 0 °C.





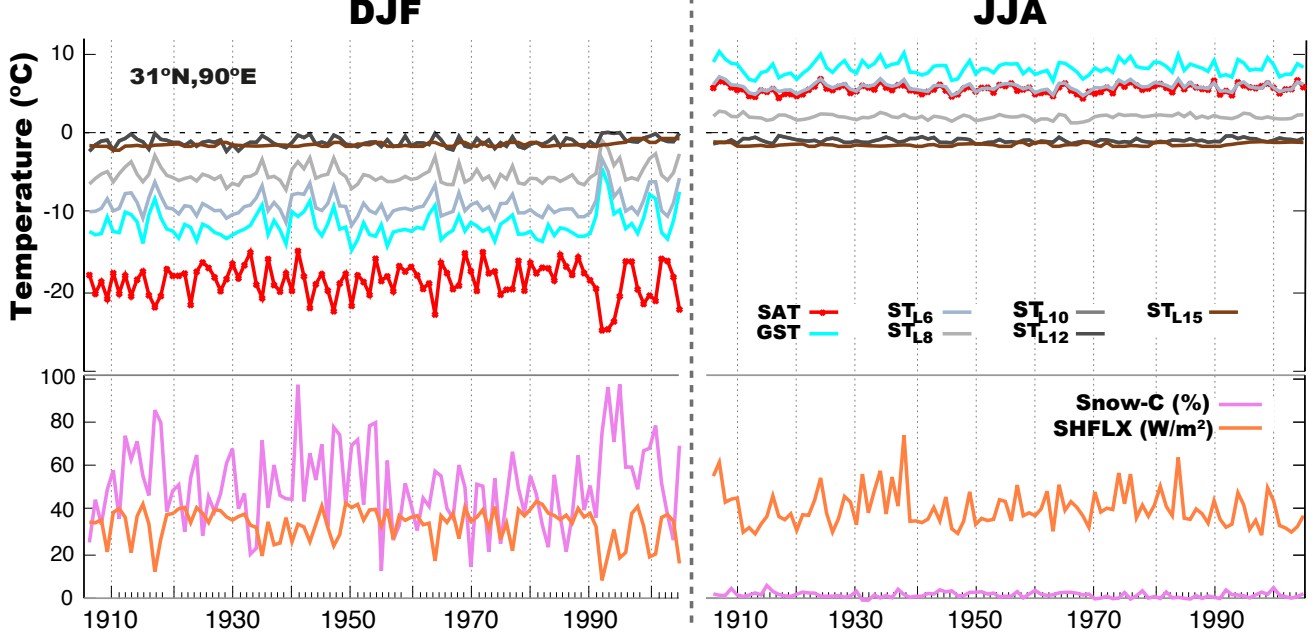

**Figure 4.** Top. DJF and JJA evolution of SAT, GST, $ST_{L6}$, $ST_{L8}$, $ST_{L10}$, $ST_{L12}$ and $ST_{L15}$ for a grid-point (Tibetan Plateau region; 31 °N, 90 °E) from the ALL-$F_2$ simulation (Table 3) during the period 1900-2005 CE. Dashed lines indicate 0 °C. Bottom. Percentage of snow cover (Snow-C) and surface sensible heat flux (SHFLX). Note that both Snow-C and SHFLX are indicated in the left axis. The units are shown within brackets in the figure legend.





**Figure 5.** Top. DJF and JJA average surface latent heat fluxes (LHFLX) over the 850-2005 CE period. Middle. DJF and JJA time evolution of SAT, GST, $ST_{L6}$, $ST_{L8}$, $ST_{L10}$, $ST_{L12}$ and $ST_{L15}$ during the period 1900-2005 CE for a grid-point (southeastern China; 30 °N, 111 °E) from the ALL-$F_2$ simulation (Table 3). Bottom. Same as middle for SHFLX, LHFLX and water content in the top 10 cm of the soil (W-10cm).





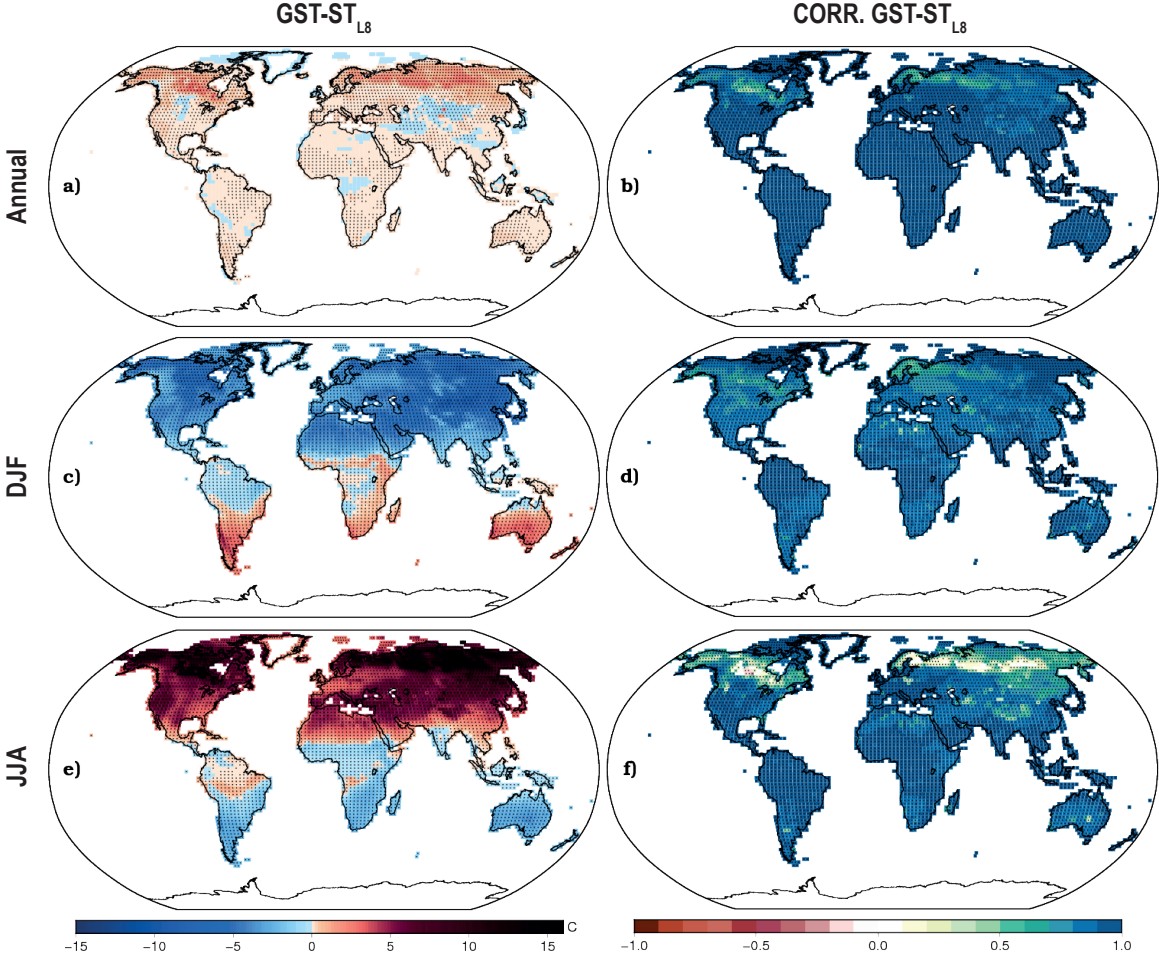

**Figure 6.** Left. (a) Annual, (c) DJF, (e) JJA spatial distribution of GST minus $ST_{L8}$ differences during the 850-2005 CE period. Right. (b) Annual, (d) DJF, (f) JJA spatial distribution of the correlation coefficients between GST and $ST_{L8}$ for the same period. Black dots indicate that 80% of the members within the ALL-F ensemble deliver statistical significance ($p<0.05$).





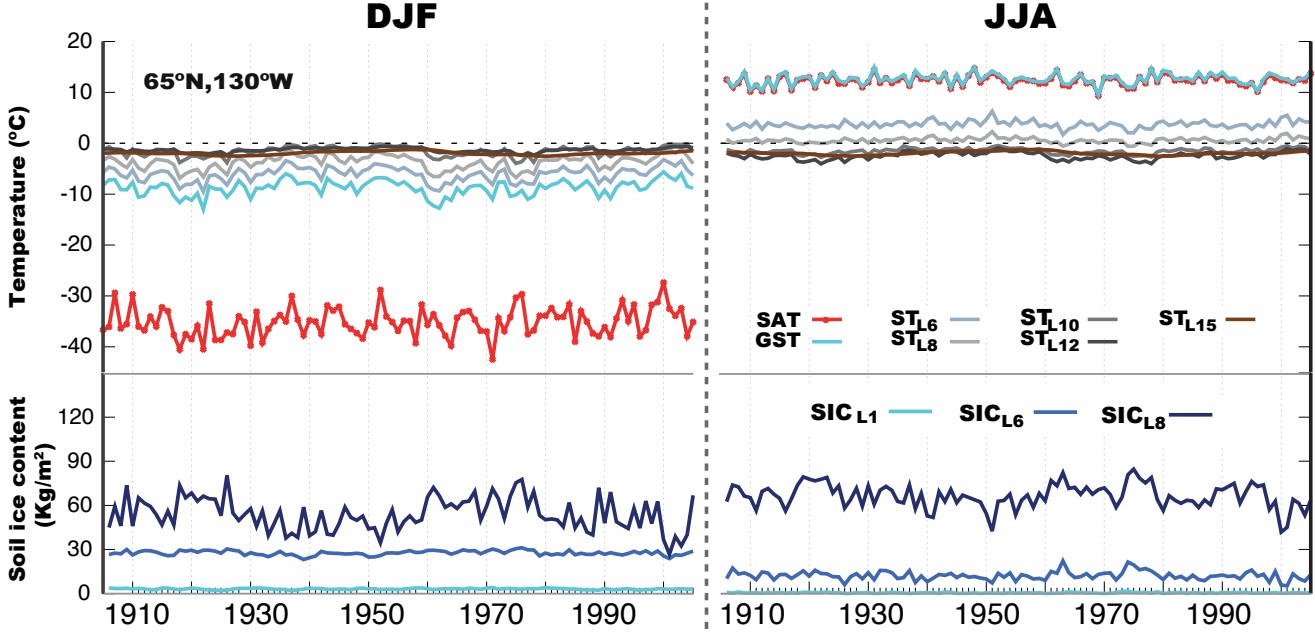

**Figure 7.** Top. DJF and JJA evolution of SAT, GST, $ST_{L6}$, $ST_{L8}$, $ST_{L10}$, $ST_{L12}$ and $ST_{L15}$ during the period 1900-2005 CE for a grid-point located at the north of Canada (65 °N, 130 °W) from the ALL-F$_2$ simulation (Table 3), an area with permanent frozen soils. Dashed lines indicate 0 °C. Bottom. Soil ice mass content (SIC) in the L1, L6 and L8 soil layers.





**Figure 8.** Spatial distribution of the linear trends in the industrial period for the ALL-F (a,b), GHG-only (c,d), LULC-only (e,f) and OZ/AER-only (g,h) ensembles for SAT (left) and GST (right). Trends are indicated in Celsius/decade. Dots indicate agreement in 80% of the ensemble members in delivering significant trends at a grid-point in the case of the ALL-F ensemble. For the GHG-only and LULC-only ensembles, dost indicate that at least two of the three ensemble members agree in delivering significant trends for a grid-point whereas for the OZ/AER-only dots indicate that both of the ensemble members deliver significant trends. Note the different scale for OZ/AER-only





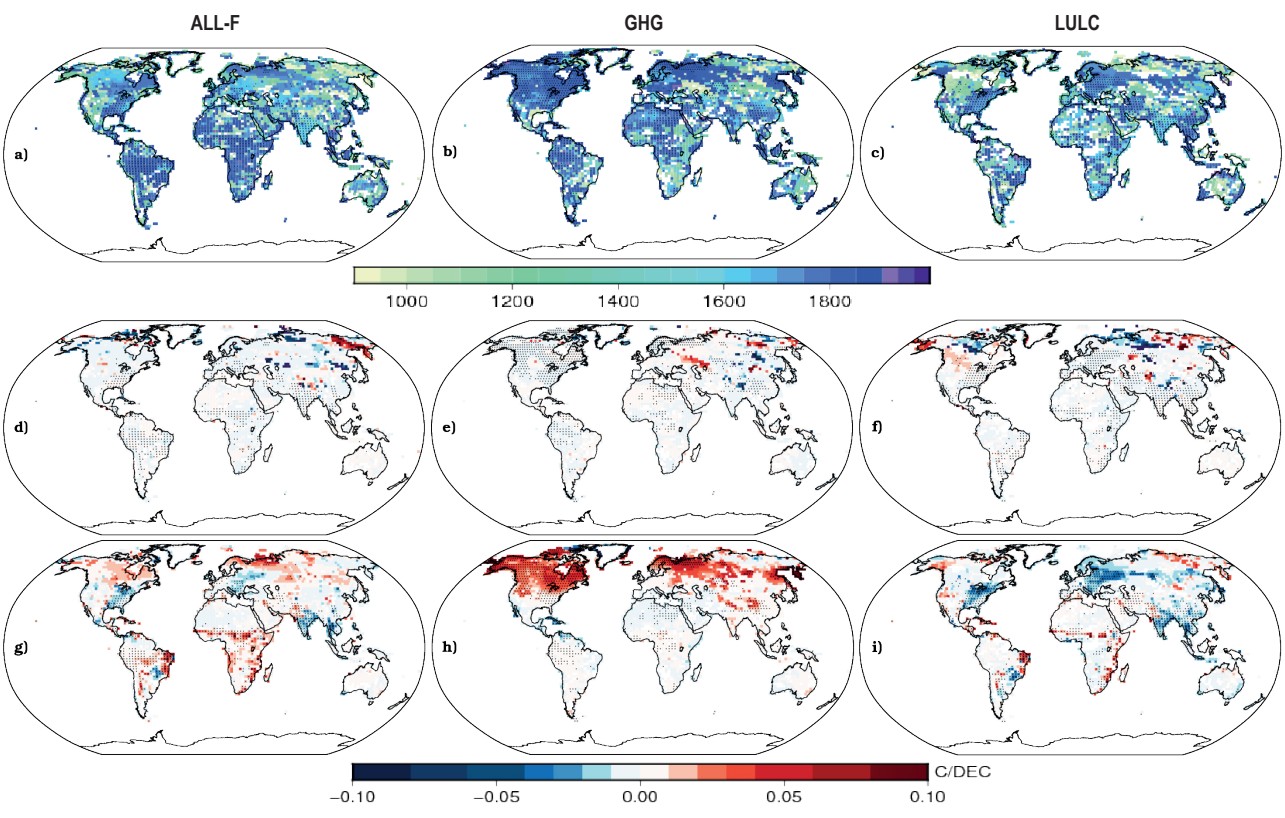

**Figure 9.** Spatial distribution of the two-phase regression model results for SAT minus GST in the ALL-F (left), GHG-only (center), LULC-only (right) ensembles: Year-of-change maps (top); trends before the change (middle); and trends after the change (bottom). Trends are indicated in Celsius/decade. Dots indicate significant level for the Year-of-change parameter based on an F-test suggesting a significant variation in the long-term trend beginning at an specific year. In the case of the ALL-F ensemble, dots indicate agreement in 80% of the ensemble members. For the GHG-only and LULC-only ensembles, dots indicate agreement in at least two of the three ensemble members





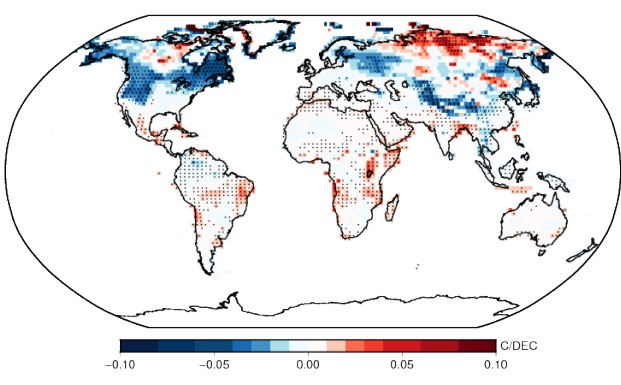

**Figure 10.** Spatial distribution of the linear trends for SAT minus GST in the OZ/AER-only ensemble. Trends are indicated in Celsius/decade. Dots indicate agreement in both of the ensemble members

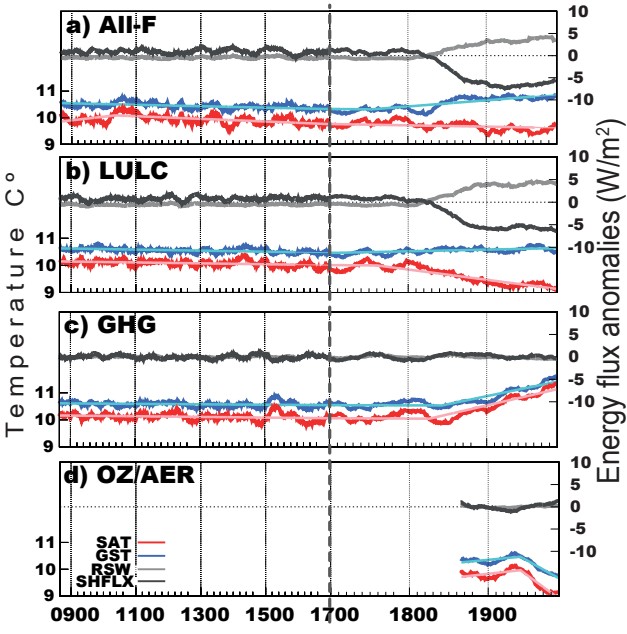

**Figure 11.** LM evolution of SAT, GST, reflected shortwave radiation (RSW) and surface sensible heat flux (SHFLX) for a grid-point located at 40 °N, 82 °W at the south of the Great Lakes, U.S. in the ALL-F$_2$, LULC$_1$, GHG$_1$ and OZ/AER$_1$ simulations (Table 3). The left axis correspond to SAT and GST while the right axis to the energy fluxes at the surface. Note that for the energy fluxes the anomalies with respect to 850-2005 CE are shown whereas for temperature absolute values are presented. All series are 31-yr moving average filter outputs. For SAT and GST the result of the two-phase regression model is displayed with thin solid lines. Note the change in timing of the x axis after 1700 CE.



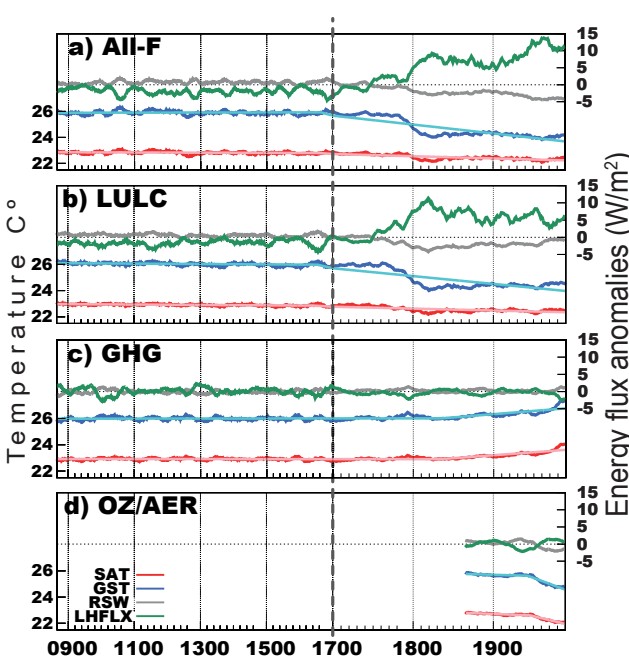

**Figure 12.** Same as in Fig. 11 but for a grid-point located at 12 °S, 40 °W at the northeast of Brazil and LHFLX ir represented instead of SHFLX.



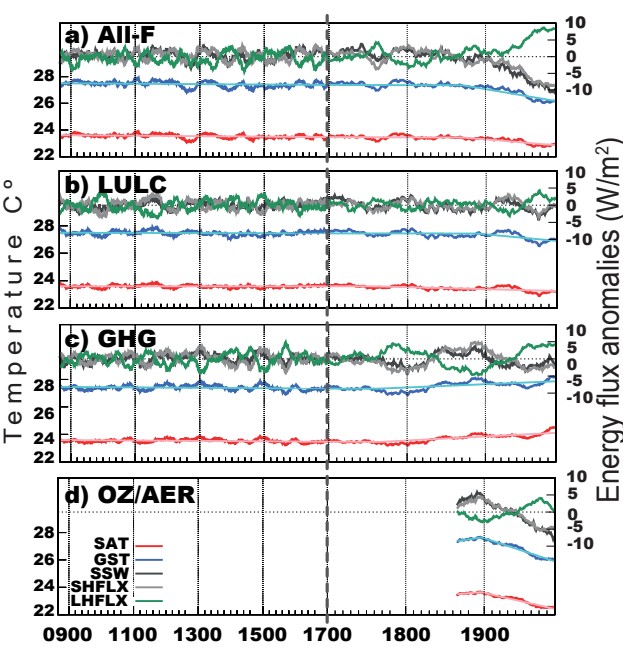

**Figure 13.** Same as in Fig. 11 but for a grid-point located at 2.5 °N, 33 °E over Uganda. Note that incoming short wave radiation at the surface (SSW) is represented instead of RSW and both LHFLX and SHFLX are shown.



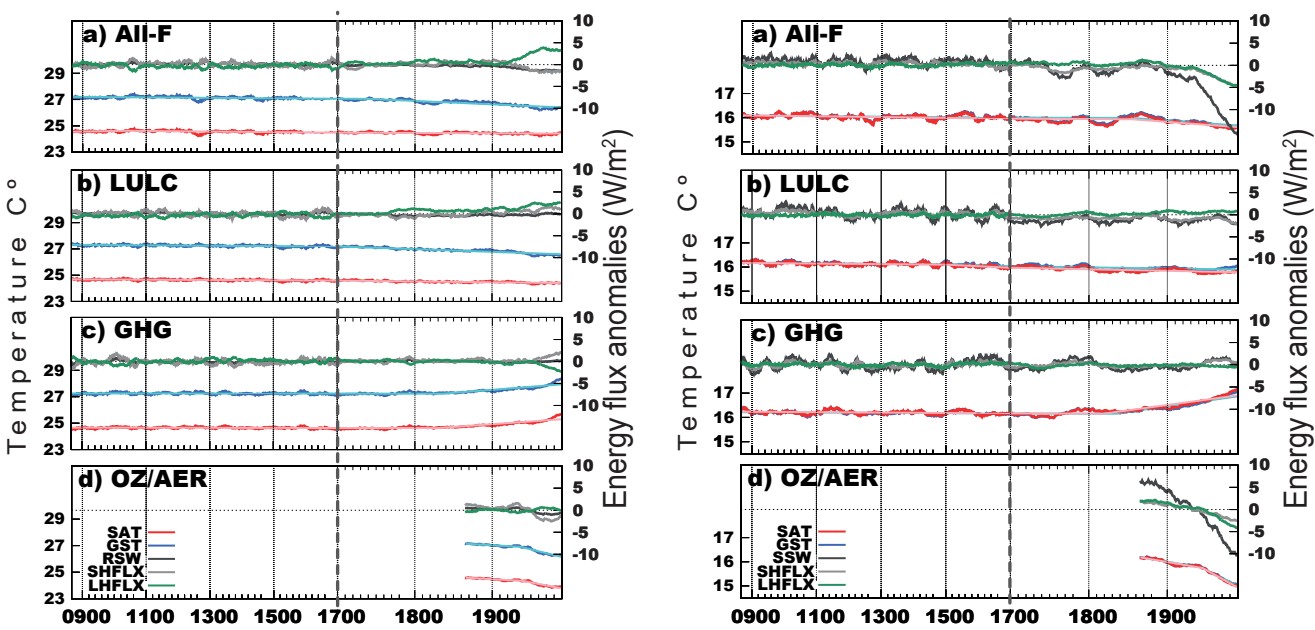

**Figure 14.** Same as in Fig. 11 but for two different regions: northeast of Brazil between 1 °S - 11 °S and 47 °W - 35 °W (left) and southeast of China between 22 °N - 32 °N and 103 °E - 122 °E (right). For the area of northeast Brazil RSW is represented whereas for the area of southeast China SSW is shown instead. LHFLX and SHFLX are shown in both cases.





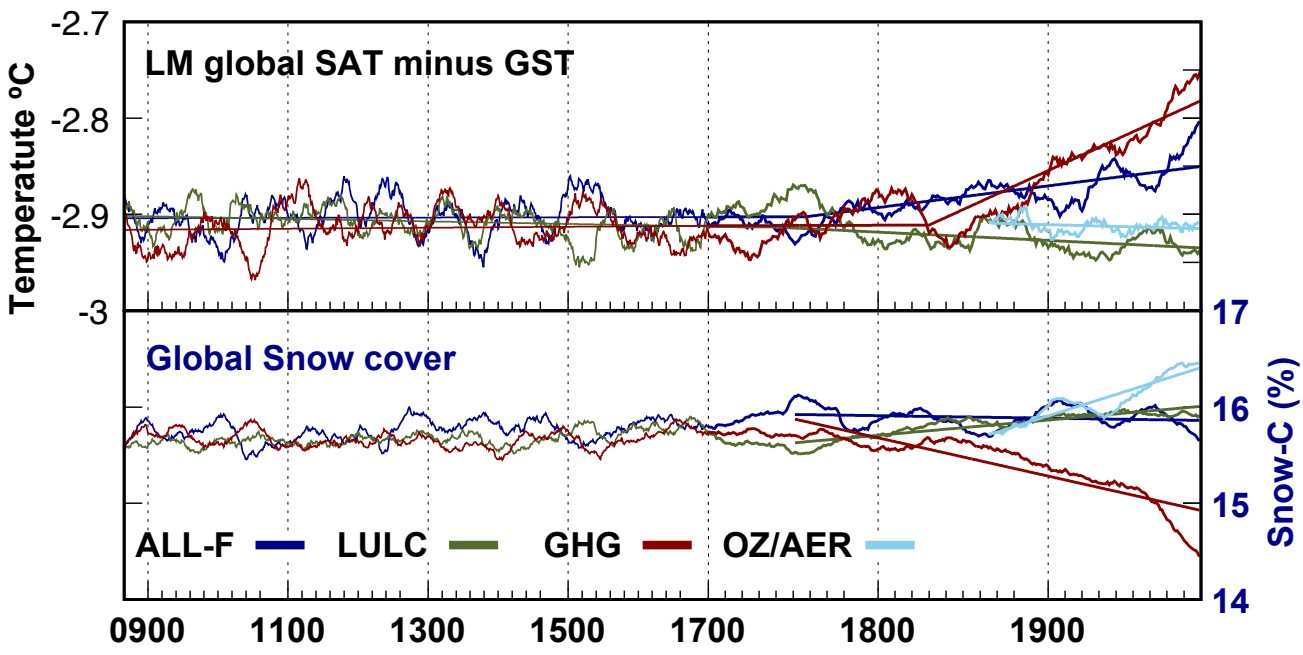

**Figure 15.** Top. LM evolution of the global SAT-GST offset for the ALL-F$_2$, LULC$_1$, GHG$_1$ and OZ/AER$_1$ simulations (Table 3). Straight lines indicate the long-term trend during the LM from a two-phase regression analysis except for the OZ/AER$_1$ that indicate linear trends in the industrial period. Bottom panel illustrates the LM evolution of global snow cover percentage for the same ensemble members. Straight lines indicate the long-term trend within the 1750-2005 period except in the case of OZ/AER that covers only the 1850-2005 period. The period from 1750 to 2005 was selected in order to match the timespan when the SAT-GST offset experiences the variation in the ALL-F$_2$ simulation. All series are 31-yr moving average filter outputs. Note the change in timing of the x axis after 1700 CE.