# Peer review of "Influence of radiative forcing factors on ground-air temperature coupling during the last millennium: implications for borehole climatology"

_Climate of the Past, 2018_

## Referee Comment (RC1) · Anonymous Referee #1 · 23 Jul 2018

The manuscript presents a rather detailed investigation of the coupling between air-surface temperature and ground temperature in a series of climate simulations over the past millennium with an Earth System model. The analysis is focused on two main aspects: their co-variability and inter-annual time scales and their relative trends over the period of anthropogenic warming. The main conclusions are that in general terms bot variables are strongly coupled, but with regional particularities. The main mechanism that modulate the coupling is snow cover. Considering the long-term trends over the past century, varying snow cover due to warming causes a change of the

coupling over time that should be taken into account when using ground temperature from boreholes to reconstruct past near surface-air temperature. Deforestation over the past century also affects the coupling between surface temperature and ground temperature.

In my view the manuscript is very well written,and I have just a few minor suggestions that the authors may want to consider.

- Sometimes, the text is a bit too wordy, with long paragraphs that pack a lot of information. This is admittedly a matter of style, but I guess that a typical reader would prefer some of the very long paragraphs broken up in smaller pieces, so that, after a first reading, the information can be located more rapidly if needed.

- The explanation of the negative correlation between air temperature and ground temperature in Tibet is intriguing, but I am not sure whether it is complete. The authors argue that the intermittent heat flux - modulated by seasonal snow cover - can cause a negative correlation between the *time derivative* of both temperatures, but does this translate in a negative correlation between the temperature anomalies themselves ? This may be true depending on the spectrum of the temperature variations. If the spectrum is white, i.e. little multidecadal variability, I can see that this explanation can be correct. However, if the temperature variations are mostly decadal, changes in the time derivative of temperature may not be enough to change the sign of the temperature anomalies. All in all, I found this mechanism interesting, but I am not completely sure that it is the sole explanation. Unfortunately, I cannot suggest new ideas.

Other minor points:

- 'Improving our knowledge of the LM climate variability' LM has been defined in the abstract, but I think not in the main text

- The second, postulate is that variations in SAT propagate downward to the subsurface through conduction (Pollack et al., 1998; Smerdon et al., 2003,

Delete comma after postulate

- 'In the present work the LM refers to the period from 850 to 2005 CE while the periods from 850 to 1850 and 1851 to 2005 CE refer to pre-industrial and industrial, respectively. 2 m air temperature is used for SAT and the first land model level..'

I my understanding, a sentence in English may not start with a number. Change to 'two-meter temperature..'

-CESM-LME supports the assumption that SAT is tightly coupled with GST at global scales and above multi-decadal scales. May be change 'above' to 'longer than multi-decadal temporal scales'

- 'the thermal difference between SAT and GST. In order to explore the influence of such processes on the global SAT-GST relationship this analysis is extended considering winter and summer seasons (DJF and JJA hereafter) independently (Fig. 1c,e).–

Specify boreal summer and boreal winter

- The correlation maps (Fig. 2b) provide a similar pattern with high and significant values in most the globe (>0.8 in regions located below 45âŮę north) and lower correlations over NH mid- and high-latitudes; especially over the east of Siberia.

'below' is here ambiguous (or even incorrect): does it mean between 45N and 90S ? or between 45N and the equator. Maybe simply say ' for northern latitudes smaller than 45 N

---

## Referee Comment (RC2) · Anonymous Referee #2 · 8 Aug 2018

Summary: This manuscript uses an ensemble of LM simulations from the NCAR CESM1.1 LME to investigate the interannual to centennial coupling between SAT and GST in both time and space. The simulations allow the authors to separate different forcing influences on the coupling and to provide some detailed physical assessments of why the coupling changes over different timescales. The biggest implications of the work relate to inversions of terrestrial borehole temperature measurements as estimates of GST changes (and their interpretation as indicators of SAT changes) over the past 500-1000 years.

[Figure]

General Remarks: This is overall a well written and interesting manuscript. It is a logical next step, given the work that some of these authors have done to test SAT-GST coupling and the implications for borehole paleoclimatology in LM simulations. It goes into detail to explain the various coupling behavior that is observed and has some important conclusions. I therefore think that it should ultimately be published, but I have significant concerns about some of the analysis as outlined below. A major revision is therefore necessary before the manuscript should be accepted for publication.

The biggest analysis issue that concerns me is the two-phase regression model. The authors do not give a reasonable justification for selecting this regression model and I think it has several significant impacts on their conclusions. In particular, the break-points influence their results in ways that can be hard to interpret. One way in which this is manifest is in their physical interpretations of the coupling influences. In Figure b, d, and f we are provided time series and regression results for SAT-GST and snow cover. The author's interpretation is that snow cover has influenced the coupling, but it appears that the breakpoint comes earlier in the SAT-GST time series than it does for snow cover. If we are to take the breakpoints as physically meaningful (and I think that should be done very cautiously), why should we conclude that snow cover is driving the decoupling when the major break in its trends comes after the breakpoint in SAT-GST? The answer is that we probably should not take the breakpoints very seriously and it is a shortcoming of the analysis that the regressions are confined to the two-phase model. This is also a problem in the comparison of trends in Figure 9. The upper panels indicate that the break points vary significantly spatially, with some locations yielding breakpoints in the 1000-1200 period and many in the 1700-1800 period. All of these trends of course end at the beginning of the 21st century, meaning that the second-period trends displayed in the middle and bottom panels of Figure 9 are taken over vastly different time periods. It is very difficult to determine how to compare such trends, many of which will be impacted by strong 20th-century changes that are either much stronger because of a later breakpoint or much weaker because of an earlier breakpoint. One might justify this approach if the breakpoints could more reliably be

interpreted, but in many cases, indeed as evidenced across the ensemble, there is a lot of variability in the determined timing of the breakpoint (look at Figure 15, for instance, and try to convince yourself that the break points in the upper panel for ALL-F or OZ/AER shouldn't be several centuries in either direction of where the breakpoints were determined to be). I therefore have a very hard time interpreting the magnitude of the trends and what they actually mean. As such, I think the two-phase regression model is both poorly justified and probably a very confusing interpretation of the data.

Related to the above, the overall magnitude of the impact is hard to parse in the context of what the authors have done. They look specifically at the 1850-2005 period and note that the overall difference in SAT and GST trends looks to be about 0.05C. First of all, this should be couched in the context of the 20th-century trend estimate from the global borehole reconstruction of $\sim$0.5C/century for the 20th century. If we take the model estimate at face value, the error introduced is therefore less 10%, given the authors' estimate is over a longer period of time (it should be reported in C/century for better comparison). But the other analyses complicate this number and I am not sure how to interpret them. The trends estimated in Figure 9 for SAT-GST are +/- 0.1C per DECADE, which is a much larger regional impact than the number determined for the global difference (I understand these regional impacts would be muted in a global average, but the numbers are complicated by my subsequent point). Part of the confusion is because of an apples-to-aardvarks comparison, namely the global trend difference is from 1850-2005, while the trends in Figure 9 are for widely different intervals determined by the breakpoint in the regression analysis. In some ways it would be much cleaner if the authors determined trends over set centuries to make sure the comparisons were consistent in time and to make the comparisons to the actual borehole reconstructions simpler. Such an analysis would have its own shortcomings, but as things stand the presentation is ambiguous on the most important point of the paper, namely if and how big a bias might be present in the borehole reconstructions based on the authors' analysis.

Another issue related to the interpretation of the authors' results in the context of the borehole reconstructions is the actual spatial sampling of the borehole database and how it may actually be impacted by the biases the authors are reporting. This is particularly the case regarding snow cover and the other cryogenic effects that the authors report for the high latitudes. While these impacts are reasonably described and there are clear differences between GST and SAT particularly in the high northern latitudes, there simply aren't that many boreholes used for the global reconstructions above about 55-60 degrees N latitude. While the biases are certainly a concern and relevant to any regional studies of high-latitude boreholes, it is not clear that the regionally very large impacts play much of a role in the existing NH and global borehole reconstructions. The authors could do a lot to quantify the effect by performing an analysis that restricts the assessment to locations where boreholes exist in the reconstruction database. I would advocate for that, but it would also add to what is already a long and detailed paper. It would suffice to simply mention this issue and \*perhaps\* provide at least some numbers on the impact of the hemispheric or global estimates if, for instance, latitudes north of 60 degrees were excluded.

Specific Comments:

The paper is generally well written, but there are language issues scattered throughout. I have noted some of those below, but the paper would be improved by a thorough evaluation of typos and language choices.

Pg. 3, Ln 9: has been supported by observations...

Pg. 4, Ln. 2: PPEs experiments is redundant

Pg. 6, Ln 1: It composes a total...

Pg. 6, Ln. 12: covariance during...

Pg. 6, Ln. 16: may help characterize a potential...

Pg. 6, Ln. 28: associations

Pg. 7, Ln 4: I don't think that the OZ/AER (ozone/aerosol) abbreviation has been defined in text.

Pg. 7, Ln 5: allows identification

Pg. 7, Ln 7: through industrial times

Pg. 7, Ln 3: I don't think the subscript notation for the ensemble member or the layer depth has been explained anywhere. It should be defined for completeness.

Pg. 7, Ln 20: While the correlations with GST and the lower depths are perhaps useful to explore, it should be explained that because of the phase shift correlations will diminish even if the system is purely conductive. All borehole inversions take into account the phase shift and this kind of time series comparison is a bit misleading. The authors should clearly point this out if they are going to leave this analysis in the manuscript. For what it is worth, some of the length and detail of the manuscript would be improved by removing these specific analyses because I do not think they add much.

Pg. 7, Ln 26: very cold air temperatures

Pg. 8, Ln 30: consider "In contrast" instead of "On the contrary"

Pg. 9, Ln 28: relationship at relatively

Pg. 10, Ln 3: during JJA consistent with

Pg. 10 Ln 8: high rates

Pg. 10, Ln 24: such partitioning determines

Pg. 10, Ln 27: because the water, relative to the continental air, is warmer in winter.

Pg. 10, Ln 33: The paragraph that ends here is an example of where things get a little too detailed. I think that this paragraph in particular could be scaled down a bit.

Pg. 11, Ln 14: during these months

Pg. 12, Lns 13-14: It is relevant that other GCMs may suggest different levels of impacts, but this seems a little strange in the context of the real world. The real question is how well the GCMs represent the real-world impacts. It would be useful here to make a statement along those lines, in connection to what consistency or disagreements across models might imply for the real world.

Pg. 12, Ln 16: trends of both temperatures independently

Pg. 13, Ln 1: Why not report pattern correlations in this discussion? A quantification of the spatial similarity would seem helpful.

Pg. 15, Ln 15: the driving mechanisms

Pg. 15, Ln 26: The authors say nothing about simulated increases in LAI here. Instead of transitions to different plant functional types, it is also possible that increases in CO2 during the 20th century drive LAI increases in the CESM that in turn impact evapotranspiration. This has been shown for the model projections and also may play a role in the historical portion of the runs that the authors are analyzing. See the following paper for some background:

https://agupubs.onlinelibrary.wiley.com/doi/abs/10.1002/2018GL077051

Pg 15, Ln 32: stable state

---

## Referee Comment (RC3) · Anonymous Referee #3 · 17 Sep 2018

Comments on "Influence of radiative forcing factors on ground-air temperature coupling during the last millennium: implications for borehole climatology" Camilo Melo-Aguilar et al.

The paper examines the relationship between surface air temperature and ground surface temperature within a set of model simulations. The idea is to use the SAT-GST correlation as a metric to examine – or as a proxy, for the ground surface energy balance. This is done with the CESM-LME model ensemble that includes a set of experiments with all forcing as well as simulations with individual forcing. It is expected

that comparison of the resulting simulations would yield information regarding the importance of each of these forcing on the SAT-GST relation over spatial scales and its temporal evolution. The paper briefly mentions that this may be important for borehole climatology based climate reconstructions.

This is a generally well-written papers, but it would benefit from a full language revision.

I have several minor clarifications that I believe need to be implemented for completeness before the paper is published.

- What do "regional" and "local scale" mean in the context of this model resolution?

- How is the SAT defined within the CESM? Is it always the surface air temperature at a height of 2 m above the bare ground? Is it defined differently over a (say) forested area? If so, what could be the potential problems for the SAT-GST correlation?

- Could the author comment on the role played by evapotranspiration on the SAT-GST metric?

- What is the justification for using the two-phase regression model?

- Why do authors use the "change points" obtained from the SAT-GST regression for the analogous analysis for snow cover? Shouldn't these snow cover "change points" be determined independently in order to see whether Is a relationship exists?

- DJF and JJA refer to northern Hemisphere seasons, but are used in the global context. There is some discussion later in the text to acknowledge these restrictions, but they should be given up front or simply restrict the analysis to the northern hemisphere; likely there would be no difference in the results.

- Regarding the discussion about the Tibetan Plateau (Fig 4) , could the authors discuss whether the CESM resolution can account for the topographic variations and whether such high elevation variability could introduce unforeseen effects on SAT-GST?

- In Figure 9, are all trends for the"post-change" interval? If so, why are the values given

in K/decade? Most change points are post 1850; there should be at least a century post change. i.e. could you give the changes in K/century?

- I am not convinced that the results from this work necessarily imply that borehole climatology reconstructions would have to be revised to account for SAT-GST uncoupling. The effects are small and given the uncertainties inherent to the estimates of the quasi-steady state of the thermal regime of the ground, these additional errors are likely very small. Perhaps a simple theoretical experiment with +/- the LUC uncertainty imposed on a typical borehole temperature profile and its resulting inversion may illustrate the effect well, if any.
* * *

---

## Author Comment (AC1) · 30 Sep 2018

**Response to the reviewers' comments**

Camilo Melo-Aguilar [1,2], J. Fidel González-Rouco [1,2], Elena García-Bustamante [3], Jorge Navarro-Montesinos [3], and Norman Steinert [1,2]

[1]Universidad Complutense de Madrid, 28040 Madrid, Spain
[2]Instituto de Geociencias, Consejo Superior de Investigaciones Cientificas-Universidad Complutense de Madrid, 28040 Madrid, Spain
[3]Centro de Investigaciones Energéticas, Medioambientales y Tecnológicas (CIEMAT), 28040 Madrid, Spain

**Correspondence:** Camilo Melo-Aguilar (camelo@ucm.es)

**1 Anonymous Referee 1**

*GENERAL COMMENTS:*

GC01: *REVIEWER'S COMMENT:*

*The manuscript presents a rather detailed investigation of the coupling between air surface temperature and ground temperature in a series of climate simulations over the past millennium with an Earth System model. The analysis is focused on two main aspects: their co-variability and inter-annual time scales and their relative trends over the period of anthropogenic warming. The main conclusions are that in general terms bot variables are strongly coupled, but with regional particularities. The main mechanism that modulate the coupling is snow cover. Considering the long-term trends over the past century, varying snow cover due to warming causes a change of the coupling over time that should be taken into account when using ground temperature from boreholes to reconstruct past near surface-air temperature. Deforestation over the past century also affects the coupling between surface temperature and ground temperature. In my view the manuscript is very well written, and I have just a few minor suggestions that the authors may want to consider.*

AUTHORS' RESPONSE:

The authors welcome the positive perspective of the reviewer on the paper. We are grateful for the reviewer's comments. Please find below our reply to your review in the discussion process of our manuscript. This reply is intended to give an overall view of the analysis we have done of your comments and the way we are going to include them in the revised manuscript. The comprehensive point-to-point response will be done in a separate document, including the original manuscript with track changes so the specific modifications can be found in the text.

GC02: *REVIEWER'S COMMENT:*

*Sometimes, the text is a bit too wordy, with long paragraphs that pack a lot of information. This is admittedly a matter of style, but I guess that a typical reader would prefer some of the very long paragraphs broken up in smaller pieces, so that, after a first reading, the information can be located more rapidly if needed.*

AUTHORS' RESPONSE:

5  A thorough revision of the text is going to be done in order to re-edit and to identify which of the long paragraphs can be broken up in smaller pieces or shortened, and the changes will be implemented in the revised manuscript. We hope that the reviewer finds the changes satisfactory and the quality of the manuscript improved.

GC03:  *REVIEWER'S COMMENT:*

*The explanation of the negative correlation between air temperature and ground temperature in Tibet is intriguing, but*
10  *I am not sure whether it is complete. The authors argue that the intermittent heat flux - modulated by seasonal snow cover - can cause a negative correlation between the time derivative of both temperatures, but does this translate in a negative correlation between the temperature anomalies themselves ? This may be true depending on the spectrum of the temperature variations. If the spectrum is white, i.e. little multidecadal variability, I can see that this explanation can be correct. However, if the temperature variations are mostly decadal, changes in the time derivative of temperature may*
15  *not be enough to change the sign of the temperature anomalies. All in all, I found this mechanism interesting, but I am not completely sure that it is the sole explanation. Unfortunately, I cannot suggest new ideas.*

AUTHORS' RESPONSE:

Our explanation of the negative correlation between air and ground surface temperature is based on annual time series. Indeed, it refers specifically to anti-correlation between the anomalies themselves. Longer air and ground temperature
20  variations, such as decadal or multi-decadal, are also anti-correlated, and intermittent heat flux modulated by seasonal snow cover is the dominant factor. A negative correlation between the time derivative of both temperatures is also present. We think that perhaps a more clear explanation of the time scales used to describe the main mechanisms impacting the SAT-GST coupling may be needed in the text. This will be included in the revised manuscript, trying not to increase the length of the text.

25  *SPECIFIC COMMENTS:*

Other minor points:

SC01  *:* 'Improving our knowledge of the LM climate variability' LM has been defined in the abstract, but I think not in the main text

30  Answer: LM is going to be defined in the main text of the revised manuscript

SC02 *:* The second, postulate is that variations in SAT propagate downward to the subsurface through conduction (Pollack et al., 1998; Smerdon et al., 2003, Delete comma after postulate

  Answer: The comma will be deleted in the revised manuscript

SC03 *:* 'In the present work the LM refers to the period from 850 to 2005 CE while the periods from 850 to 1850 and 1851 to 2005 CE refer to pre-industrial and industrial, respectively. 2 m air temperature is used for SAT and the first land model level..' I my understanding, a sentence in English may not start with a number. Change to 'two-meter temperature..'

  Answer: The text will be changed as suggested in the revised manuscript

SC04 *:* CESM-LME supports the assumption that SAT is tightly coupled with GST at global scales and above multi-decadal scales. May be change 'above' to 'longer than multidecadal temporal scales'

  Answer: This will be changed as suggested in the revised manuscript

SC05 *:* 'the thermal difference between SAT and GST. In order to explore the influence of such processes on the global SAT-GST relationship this analysis is extended considering winter and summer seasons (DJF and JJA hereafter) independently (Fig. 1c,e).- Specify boreal summer and boreal winter

  Answer: This will be changed as suggested in the revised manuscript

SC06 *:* The correlation maps (Fig. 2b) provide a similar pattern with high and significant values in most the globe (>0.8 in regions located below 45° north) and lower correlations over NH mid- and high-latitudes; especially over the east of Siberia. 'below' is here ambiguous (or even incorrect): does it mean between 45N and 90S ? or between 45N and the equator. Maybe simply say 'for northern latitudes smaller than 45 N

  Answer: The reviewer is right in pointing out that this may be ambiguous. Actually, "below" here refers to land areas between 45N and 90S. This will be changed in the revised manuscript to "regions located between 45° N and 90° S.

**2 Anonymous Referee 2**

*GENERAL COMMENTS:*

5 GC01: *REVIEWER'S COMMENT:*

*This is overall a well written and interesting manuscript. It is a logical next step, given the work that some of these authors have done to test SATGST coupling and the implications for borehole paleoclimatology in LM simulations. It goes into detail to explain the various coupling behavior that is observed and has some important conclusions. I therefore think that it should ultimately be published, but I have significant concerns about some of the analysis as outlined below.*

10 *A major revision is therefore necessary before the manuscript should be accepted for publication.*

AUTHORS' RESPONSE:

We appreciate the positive perspective on the manuscript as well as the comments and suggestions. We are considering each comment of the reviewer and we are sure this will help improving the next version of the manuscript.

Please find below our reply to your review. This reply is intended to give an overall view of the analysis we have done

15 of your comments and the way we are going to include them in the revised manuscript. The comprehensive point-to-point response will be done in a separate document, including the original manuscript with track changes so the specific modifications can be found in the text.

GC02: *REVIEWER'S COMMENT:*

*The biggest analysis issue that concerns me is the two-phase regression model. The authors do not give a reasonable*

20 *justification for selecting this regression model and I think it has several significant impacts on their conclusions. In particular, the breakpoints influence their results in ways that can be hard to interpret. One way in which this is manifest is in their physical interpretations of the coupling influences. In Figure b, d, and f we are provided time series and regression results for SAT-GST and snow cover. The author's interpretation is that snow cover has influenced the coupling, but it appears that the breakpoint comes earlier in the SAT-GST time series than it does for snow cover. If we are to take*

25 *the breakpoints as physically meaningful (and I think that should be done very cautiously), why should we conclude that snow cover is driving the decoupling when the major break in its trends comes after the breakpoint in SAT-GST? The answer is that we probably should not take the breakpoints very seriously and it is a shortcoming of the analysis that the regressions are confined to the two-phase model.*

AUTHORS' RESPONSE:

30 We agree that improving the description of the two-phase regression and its use in this paper will be good for the text. Reviewer #3 has also shown some concern about this. We will include a more detailed motivation in Sect 3. We think that a better description of the physical interpretation as well as for the timing of the changes is necessary so it will be included in the revised document.

Figure 1b, d, and f illustrates that globally, snow cover is the main mechanism determining the SAT-GST offset during the LM. However, there is not an actual breakpoint in snow cover since it remains relatively stable during the LM. Thus, the breakpoint in SAT-GST cannot be solely explained by the influence of snow cover. We used the breakpoint in SAT-GST as a reference for depicting the linear regression of snow cover, for the same time periods, in order to show that snow cover does not vary at the time when SAT-GST does. Hence, some other mechanisms should be influencing the long-term variation in the SAT-GST offset. With this in mind, we explored different mechanisms in the subsequent Sections for explaining the breakpoint in SAT-GST

We think that the description of this argument needs to be improved. Therefore, we will revise how it is presented, and the required changes will be included in the revised document.

GC03: *REVIEWER'S COMMENT:*

*This is also a problem in the comparison of trends in Figure 9. The upper panels indicate that the break points vary significantly spatially, with some locations yielding breakpoints in the 1000-1200 period and many in the 1700-1800 period. All of these trends of course end at the beginning of the 21st century, meaning that the second-period trends displayed in the middle and bottom panels of Figure 9 are taken over vastly different time periods. It is very difficult to determine how to compare such trends, many of which will be impacted by strong 20th-century changes that are either much stronger because of a later breakpoint or much weaker because of an earlier breakpoint. One might justify this approach if the breakpoints could more reliably be interpreted, but in many cases, indeed as evidenced across the ensemble, there is a lot of variability in the determined timing of the breakpoint (look at Figure 15, for instance, and try to convince yourself that the break points in the upper panel for ALL-F or OZ/AER shouldn't be several centuries in either direction of where the breakpoints were determined to be). I therefore have a very hard time interpreting the magnitude of the trends and what they actually mean. As such, I think the two-phase regression model is both poorly justified and probably a very confusing interpretation of the data.*

AUTHORS' RESPONSE:

This is an interesting point raised by the reviewer. There is actually a large spatial variability of the breakpoints depicted in Fig. 9. It is important to bear in mind that only the breakpoints delivering statistically significance indicate an actual change in the long-term SAT-GST relationship. These points are mainly concentrated within the 19th century, with some exceptions that are indicated at earlier times as a response of land use land cover changes occurring before the beginning of the industrial period. This issue is discussed in Sect. 4.1 (page 13, line 24-30 of the submitted manuscript). According to this, the trends that should be considered refer consistently to the same time period.

We think that this argument may be better depicted in Fig. 9. For instance, by considering only the grid-points where significant variations in the long-term SAT-GST relationship are present. In that way, the distribution of those points around the 19th century would be visible and the comparison of such trends could be done directly. We will evaluate the

best way of presenting this information reordering both the related text and figures and the changes will be included in the revised manuscript.

GC04: *REVIEWER'S COMMENT:*

*Related to the above, the overall magnitude of the impact is hard to parse in the context of what the authors have done. They look specifically at the 1850-2005 period and note that the overall difference in SAT and GST trends looks to be about 0.05C. First of all, this should be couched in the context of the 20th-century trend estimate from the global borehole reconstruction of ~0.5C/century for the 20th century. If we take the model estimate at face value, the error introduced is therefore less 10%, given the authors' estimate is over a longer period of time (it should be reported in C/century for better comparison). But the other analyses complicate this number and I am not sure how to interpret them. The trends estimated in Figure 9 for SAT-GST are +/- 0.1C per DECADE, which is a much larger regional impact than the number determined for the global difference (I understand these regional impacts would be muted in a global average, but the numbers are complicated by my subsequent point). Part of the confusion is because of an apples-to-aardvarks comparison, namely the global trend difference is from 1850-2005, while the trends in Figure 9 are for widely different intervals determined by the breakpoint in the regression analysis. In some ways it would be much cleaner if the authors determined trends over set centuries to make sure the comparisons were consistent in time and to make the comparisons to the actual borehole reconstructions simpler. Such an analysis would have its own shortcomings, but as things stand the presentation is ambiguous on the most important point of the paper, namely if and how big a bias might be present in the borehole reconstructions based on the authors' analysis.*

AUTHORS' RESPONSE:

As discussed in the previous point (GC03), the bulk of the significant variations take place within the 19th century. We will consider changing Fig. 9 to better illustrate this fact. According to this, the trends in Fig. 9 that actually should be considered are consistent with the time interval of the global trend.

The use of a fixed time period, as for instance from 1850 to 2005, to determine the trends as the reviewer suggests is indeed an interesting approach that we also considered. On the other hand, allowing for the trends to optimally show when changes related to each forcing occurs, provides arguably more complete information and of interest for a broader community. We are not aware of any text, and there may be few, having shows the temperature changes in response to each forcing within decadal timescales and in the context of the last millennium. Results also express the spatial dependence of the response without imposing bounds. Nevertheless, we will reconsider both approaches and evaluate the best way of presenting the results. We think that by improving this, the text will gain clarity

GC05: *REVIEWER'S COMMENT:*

*Another issue related to the interpretation of the authors' results in the context of the borehole reconstructions is the actual spatial sampling of the borehole database and how it may actually be impacted by the biases the authors are reporting. This is particularly the case regarding snow cover and the other cryogenic effects that the authors report for*

*the high latitudes. While these impacts are reasonably described and there are clear differences between GST and SAT particularly in the high northern latitudes, there simply aren't that many boreholes used for the global reconstructions above about 55-60 degrees N latitude.*

*While the biases are certainly a concern and relevant to any regional studies of high-latitude boreholes, it is not clear that the regionally very large impacts play much of a role in the existing NH and global borehole reconstructions. The authors could do a lot to quantify the effect by performing an analysis that restricts the assessment to locations where boreholes exist in the reconstruction database. I would advocate for that, but it would also add to what is already a long and detailed paper. It would suffice to simply mention this issue and \*perhaps\* provide at least some numbers on the impact of the hemispheric or global estimates if, for instance, latitudes north of 60 degrees were excluded.*

AUTHORS' RESPONSE:

The reviewer raises a very interesting point that we have been considering. However, the analysis of the impacts that this kind of long-term SAT-GST decoupling processes may have on the actual spatial distribution of the borehole dataset goes beyond the scope of the present work. As the reviewer states, this manuscript is already long and detailed. Additionally, its main purpose is to give an overall view of the different processes that may play a role in impacting the long-term SAT-GST coupling assumption. With this in mind, we are currently working on a paper that aims to explore in detail the impacts on borehole reconstructions in the context of the actual borehole distribution. Thus, the specific numbers will be addressed in subsequent work.

**SPECIFIC COMMENTS:**

The paper is generally well written, but there are language issues scattered throughout. I have noted some of those below, but the paper would be improved by a thorough evaluation of typos and language choices.

SC01  *Pg. 3, ln 9:* has been supported by observations...

  Answer: This will be corrected in the revised manuscript.

SC02  *Pg 4, ln 2:* PPEs experiments is redundant

  Answer: We agree with the reviewer. Therefore the redundant expression will be deleted in the revised manuscript.

SC03  *Pg 6, ln 1:* It composes a total...

  Answer: The text will be changed as suggested in the revised manuscript

SC04  *Pg 6, ln 12:* covariance during...

  Answer: This will be corrected in the revised manuscript

SC05 *Pg 6, ln 16:* may help characterize a potential...

      Answer: The text will be changed as suggested in the revised manuscript

SC06 *Pg 6, ln 28:* associations

      Answer: The expression refers to a singular form therefore it will not be changed.

SC07 *Pg 7, ln 4:* I don't think that the OZ/AER (ozone/aerosol) abbreviation has been defined in the text.

      Answer: It is defined in Table 2. which is mentioned in Section 2. However, we will check whether it needs to be defined in the main text. The changes will be included in the revised manuscript.

SC08 *Pg 7, ln 5:* allows identification

      Answer: The text will be changed in the revised manuscript.

SC09 *Pg 7, ln 7:* through industrial times

      Answer: This will be corrected in the revised manuscript.

SC10 *Pg 7, ln 3:* I don't think the subscript notation for the ensemble member or the layer depth has been explained anywhere. It should be defined for completeness.

      Answer: The subscript notation for the ensemble member is explained in Table 3. and it is mentioned in Sect. 2. page 6, line 4 of the submitted document. The subscript notation for the layer depth is explained in Table 1. However, we think that this could be specifically mentioned in the main document, specifically in Section 3. Therefore, a brief explanation will be included in the revised manuscript

SC11 *Pg 7, ln 20:* While the correlations with GST and the lower depths are perhaps useful to explore, it should be explained that because of the phase shift correlations will diminish even if the system is purely conductive. All borehole inversions take into account the phase shift and this kind of time series comparison is a bit misleading. The authors should clearly point this out if they are going to leave this analysis in the manuscript. For what it is worth, some of the length and detail of the manuscript would be improved by removing these specific analyses because I do not think they add much.

      Answer: Indeed the reviewer is right in pointing out the fact that the phase shift of the downward propagating signal diminishes the correlation between GST and temperature at lower depths even if the system is purely conductive. This may be the general argument by which depth impacts correlations by amplitude damping and phase shift. Actually, we stressed this issue in the first part of Sect. 4, (page 7, line 21) where the discussion about the correlation between SAT and ST at different depths is presented. Nevertheless, Fig. 6 shows that, in spite of the phase shift, correlations within the upper meter of the subsurface are large in general over the areas where the regime is purely conductive. Additionally, Fig. 6 clearly shows that the covariance of the signal weakens and even disappears during the boreal summer over the northernmost part of the globe where the regime is not purely conductive. We consider that this result deserves being shown in the manuscript because it gives some relevant information about the influence of non-conductive processes on

the subsurface heat transfer and was discussed in page 11, line 15-35 of the submitted manuscript. In addition, Fig 6. gives a spatial view of the amplitude attenuation with depth of the temperature signal at annual time scales that may be of interest for some readers. Nevertheless, we agree that some clarification may be needed in the text as the reviewer points out. Therefore, this part will be revised and the modifications will be included in the revised manuscript.

SC12 *Pg 7, ln 26:* very cold air temperatures

Answer: The text will be changed in the revised manuscript.

SC13 *Pg 8, ln 30:* consider "In contrast" instead of "On the contrary"

Answer: It will be replaced as suggested in the revised manuscript.

SC14 *Pg 9, ln 28:* relationship at relatively

Answer: The text will be corrected in the revised manuscript.

SC15 *Pg 10, ln 3:* during JJA consistent with

Answer: It will be corrected in the revised manuscript.

SC16 *Pg 10, ln 8:* high rates

Answer: This will be corrected in the revised manuscript.

SC17 *Pg 10, ln 24:* such partitioning determines

Answer: It will be corrected in the revised manuscript.

SC18 *Pg 10, ln 27:* because the water, relative to the continental air, is warmer in winter.

Answer: The text will be corrected in the revised manuscript.

SC19 *Pg 10, ln 33:* The paragraph that ends here is an example of where things get a little too detailed. I think that this paragraph in particular could be scaled down a bit.

Answer: We agree. A thorough revision of the text will be done in order to identify also other paragraphs that can be shortened. The changes will be implemented in the revised manuscript.

SC20 *Pg 11, ln 14:* during these months

Answer: Thanks, the text will be corrected in the revised manuscript.

SC21 *Pg 12, lns 13-14:* It is relevant that other GCMs may suggest different levels of impacts, but this seems a little strange in the context of the real world. The real question is how well the GCMs represent the real-world impacts. It would be useful here to make a statement along those lines, in connection to what consistency or disagreements across models might imply for the real world.

Answer: It is indeed true that different GCMs could indicate different level of impacts in the SAT-GST relationship during the LM. This difference would depends on how different land surface models represent the physical processes at the surface as snow cover, and soil moisture among others. For the shorter timescales even internal variability will play a role.

In spite of the possible differences, the overall impact may be expected to be consistent among the available simulations of the LM with different GCMs, since the external forcings considered in the PMIP3/CMIP5 LM simulations are similar (Schmidt et al. , 2011) and should have a similar contribution. Nevertheless, not all model simulations consider exactly the same set of forcings. Some consider land use land cover changes, and aerosols, and other don't (Masson-Delmotte et al. , 2013). This will arguably have some more profound impacts as shown in our manuscript. Even if the role of external forcings should be dominant on a long-term context, Addressing this issues from a multi-model ensemble approach would help understanding the uncertainties associated to all the factors of variations describe above.

This work describes the most relevant physical mechanisms and feedbacks that may affect SAT-GST coupling in different parts of the world, many of which have not yet been measure systematically, and how these processes interact with long-term changes in external forcing factors. For this purpose we have used the largest and only ensemble of model simulations including several all-forcing and single forcing realizations. We would expect that, largest ensembles with a more systematic samplings of forcings and processes will be available in CMIP6/PMIP4 (Eyring et al. , 2017; Jungclaus et al. , 2017). This will allows for exercises exploring better sensitivity to different models.

We will consider a brief discussion on this issue and the required changes will be included in the revised manuscript.

SC22 *Pg 12, ln 16:* trends of both temperatures independently

Answer: The text will be corrected in the revised manuscript.

SC23 *Pg 13, ln 1:* Why not report pattern correlations in this discussion? A quantification of the spatial similarity would seem helpful.

Answer: The pattern correlation will be included as a measure of the spatial similarity

SC24 *Pg 15, ln 15:* the driving mechanisms

Answer: Thanks, the text will be corrected in the revised manuscript.

SC25 *Pg 15, ln 26:* The authors say nothing about simulated increases in LAI here. Instead of transitions to different plant functional types, it is also possible that increases in CO2 during the 20th century drive LAI increases in the CESM that in turn impact evapotranspiration. This has been shown for the model projections and also may play a role in the historical portion of the runs that the authors are analyzing. See the following paper for some background: https://agupubs.onlinelibrary.wiley.com/doi/abs/10.1002/2018GL077051

Answer: The reviewer is right in pointing out the role of increased LAI on increasing evapotranspiration rates. It is indeed very interesting that increases in LAI, as a response to enriched $CO_2$ environment, lead to increases in

evapotranspiration, and possibly to long-term variations in the SAT-GST relationship. Actually, the Community land model version 4 (CLM4), the land component included in the CEMS-LME, incorporates a Carbon-Nitrogen module. We will address this issue of the potential effects on LAI and report back to the reviewer in the final response.

5   SC26   *Pg 15, ln 32:* stable state

> Answer: The text will be corrected in the revised manuscript.

**3   Anonymous Referee 3**

*GENERAL COMMENTS:*

5 GC01: *REVIEWER'S COMMENT:*

*The paper examines the relationship between surface air temperature and ground surface temperature within a set of model simulations. The idea is to use the SAT-GST correlation as a metric to examine – or as a proxy, for the ground surface energy balance. This is done with the CESM-LME model ensemble that includes a set of experiments with all forcing as well as simulations with individual forcing. It is expected that comparison of the resulting simulations would*

10 *yield information regarding the importance of each of these forcing on the SAT-GST relation over spatial scales and its temporal evolution. The paper briefly mentions that this may be important for borehole climatology based climate reconstructions.*

*This is a generally well-written papers, but it would benefit from a full language revision.*

*I have several minor clarifications that I believe need to be implemented for completeness before the paper is published.*

15   AUTHORS' RESPONSE:

We appreciate the positive perspective on the manuscript as well as the comments and suggestions. We are considering each comment of the reviewer and we are sure this will help improving the next version of the manuscript.

Please find below our reply to your review. This reply is intended to give an overall view of the analysis we have done of your comments and the way we are going to include them in the revised manuscript. The comprehensive point-to-

20   point response will be done in a separate document, including the original manuscript with track changes so the specific modifications can be found in the text.

GC02: *REVIEWER'S COMMENT:*

*What do "regional" and "local scale" mean in the context of this model resolution?*

AUTHORS' RESPONSE:

25   Considering the horizontal resolution of the CESM-LME ~2°, in this work local scale refers to the scale of resolution a single or a few grid points while for the regional scale the range between $10^4$ to $10^7$ km$^2$ can be considered as defined in (Giorgi et al. , 2001). From this perspective it is really broad and refers in a general sense to sub-continental structures that can be as small as the province of Madrid and as large as the U.S. We will consider the use of this concepts and make sure that it is senseful in the revised text.

30 GC03: *REVIEWER'S COMMENT:*

*How is the SAT defined within the CESM? Is it always the surface air temperature at a height of 2 m above the bare ground? Is it defined differently over a (say) forested area? If so, what could be the potential problems for the SAT-GST correlation?*

AUTHORS' RESPONSE:

The variable we used for SAT is the temperature near the surface or 2-m air temperature identifyed by the model output "TREFHT".

TREFHT is calculated by interpolation between the surface temperature and the lowest atmosphere model level temperature. The surface temperature depends on the energy fluxes at the surface. For non-vegetated areas, the surface temperature is obtained from the energy balance at the ground surface. For vegetated surfaces, the energy fluxes are partitioned into vegetation and ground fluxes (Oleson et al. , 2010). Therefore, the surface temperature is the result of the balance between both the vegetation and the ground fluxes.

Since in both cases the energy fluxes from the different surface components are considered in the calculations of the surface temperature, and consequently on the 2-m air temperature, there should not be "potential problems" for the SAT-GST correlation.

From the surrogate reality of the GCMs simulated air temperature represents the best available alternative for exploring the air and ground temperature relationship. Previous works of this type have also used 2-m air temperature from different GCMs for investigating the covariance structure between the air and ground temperatures (e.g. González-Rouco et al. , 2009; García-García et al. , 2009).

GC04: *REVIEWER'S COMMENT:*

*Could the author comment on the role played by evapotranspiration on the SAT-GST metric?*

AUTHORS' RESPONSE:

Evapotranspiration plays an important role on heat transfer from the surface to the atmosphere. It is involved in the energy balance at the surface. However, evapotranspiration does not necessarily influence the SAT-GST metric by itself. The SAT-GST long-term relationship may be affected if the means by which heat is transfered from the surface to the atmosphere experiences a long-term variations. There are a couple of examples in the document describing how long-term variations in evapotranspiration, as a response to changes in the surface characteristics, lead to long-term changes in the SAT-GST metric. For instance in Fig. 12 of the submitted manuscript it is presented how a significant increase in evapotranspiration impacts the SAT-GST relationship.

On the contrary, if the energy fluxes at the surface are relatively stable in time, the SAT-GST metric would remain also stable. We will consider including comments of this topic in the revised text.

GC05: *REVIEWER'S COMMENT:*

*What is the justification for using the two-phase regression model?*

AUTHORS' RESPONSE:

This comment is also raised by anonymous referee #2. A more detailed motivation will be included in the revised manuscript.

GC06: *REVIEWER'S COMMENT:*

*Why do authors use the "change points" obtained from the SAT-GST regression for the analogous analysis for snow cover? Shouldn't these snow cover "change points" be determined independently in order to see whether Is a relationship exists?*

AUTHORS' RESPONSE:

This is actually a point that also concerns anonymous referee #2. The change points from the SAT-GST regression were used as a reference for snow cover because there is not a significant change in the global snow cover. Thus, the use of the change points from the SAT-GST allows for exposing that besides snow cover there should be other mechanisms influencing the long-term variation in the SAT-GST offset. With this in mind, we explore different mechanisms that may be influencing such long-term variation, as is presented in the subsequent Sections of the manuscript.

As it is a common concern for the reviewers, we think that the description of this argument needs to be improved. Therefore, we will revise the how it is presented. The required changes will be included in the revised manuscript.

GC07: *REVIEWER'S COMMENT:*

*DJF and JJA refer to northern Hemisphere seasons, but are used in the global context. There is some discussion later in the text to acknowledge these restrictions, but they should be given up front or simply restrict the analysis to the northern hemisphere; likely there would be no difference in the results.*

AUTHORS' RESPONSE:

This will be revised to clarify it from the beginning of the document that DJF and JJA refer explicitly to the northern hemisphere seasons. The modifications will be included in the revised manuscript.

GC08: *REVIEWER'S COMMENT:*

*Regarding the discussion about the Tibetan Plateau (Fig 4) , could the authors discuss whether the CESM resolution can account for the topographic variations and whether such high elevation variability could introduce unforeseen effects on SAT-GST?*

AUTHORS' RESPONSE:

The topographic variations over complex mountainous regions are roughly represented in most state-of-the-art GCMs due to its relative coarse horizontal resolution. In the case of the CESM-LME, the elevation of each grid point is considered as the mean elevation for the whole area of the grid point. Thus, the orographic variations are smoothed out.

The limited characterization of the topography in most GCMs, in addition to the sub-grid-scale parameterizations, does not allow for an accurate representation of some hydrological processes over complex terrain (Huning & Margulis , 2018; Rhoades et al. , 2018). This situation may indeed lead to unforeseen effects on the SAT-GST relationship since the simulated snow cover fraction plays an important role on the effect described over the Tibetan Plateau in the CESM-LME.

Consequently, a brief mention on this issue deserves being included in the manuscript, since the specific effect on the SAT-GST relationship over complex mountainous regions could be model-dependent. We will consider this and the required changes will be included.

5 GC09: *REVIEWER'S COMMENT:*

*In Figure 9, are all trends for the"post-change" interval? If so, why are the values given in K/decade? Most change points are post 1850; there should be at least a century post change. i.e. could you give the changes in K/century?*

AUTHORS' RESPONSE:

The trends will be indicated in K/century in the revised manuscript as suggested.

10 GC010: *REVIEWER'S COMMENT:*

*I am not convinced that the results from this work necessarily imply that borehole climatology reconstructions would have to be revised to account for SAT-GST uncoupling. The effects are small and given the uncertainties inherent to the estimates of the quasi-steady state of the thermal regime of the ground, these additional errors are likely very small. Perhaps a simple theoretical experiment with +/- the LUC uncertainty imposed on a typical borehole temperature profile*

15 *and its resulting inversion may illustrate the effect well, if any.*

AUTHORS' RESPONSE:

This is an interesting point. Actually the effects that the results of this work could have on borehole reconstructions are being explored in a subsequent work. At global scales the effects resulting from this analysis are minor. At regional and local, they may not be so.

**References**

Eyring, V., Bony, S., Meehl, G. A., Senior, C. A., Stevens, B., Stouffer, R. J., Taylor, K. E.,: Overview of the Coupled Model Intercomparison Project Phase 6 (CMIP6) experimental design and organization, Geoscientific Model Development, 9, 5,1937–1958, 10.5194/gmd-9-1937-2016, 2016

García-García, A., Cuesta-Valero, F. J., Beltrami, H, Smerdon, J. E., Simulation of air and ground temperatures in PMIP3/CMIP5 last millennium simulations: implications for climate reconstructions from borehole temperature profiles, Environmental Research Letters, 11, 4, 044–022, 2016

Giorgi F., and Coauthors: Regional Climate Information-Evaluation and Projections In J.T. Houghton et al (eds.):, Climate change 2001: The Scientific Basis, Cambridge University Press, 583-638 , Cambridge, United Kingdom and New York, NY, USA.

González-Rouco, J. F., Beltrami, H., Zorita, E., Stevens, B.,: Borehole climatology: a discussion based on contributions from climate modelling, Clim. Past, 5, 99-127, 2009

Huning, Laurie S. and Margulis, Steven A., Investigating the Variability of High-Elevation Seasonal Orographic Snowfall Enhancement and Its Drivers across Sierra Nevada, California,Journal of Hydrometeorology,19(1),47-67,2018,https://doi.org/10.1175/JHM-D-16-0254.1

Jungclaus, J. H., Bard, E., Baroni, M., Braconnot, P., Cao, J., Chini, L. P., Egorova, T., Evans, M., González-Rouco, J. F., Goosse, H., Hurtt, G. C., Joos, F., Kaplan, J. O., Khodri, M., Klein Goldewijk, K., Krivova, N., LeGrande, A. N., Lorenz, S. J., Luterbacher, J., Man, W., Maycock, A. C., Meinshausen, M., Moberg, A., Muscheler, R., Nehrbass-Ahles, C., Otto-Bliesner, B. I., Phipps, S. J., Pongratz, J., Rozanov, E., Schmidt, G. A., Schmidt, H., Schmutz, W., Schurer, A., Shapiro, A. I., Sigl, M., Smerdon, J. E., Solanki, S. K., Timmreck, C., Toohey, M., Usoskin, I. G., Wagner, S., Wu, C.-J., Yeo, K. L., Zanchettin, D., Zhang, Q., Zorita, E.,: The PMIP4 contribution to CMIP6 – Part 3: The last millennium, scientific objective, and experimental design for the PMIP4 *past1000* simulations, Geoscientific Model Development, 10-11, 4005–4033, 10.5194/gmd-10-4005-2017, 2017

Masson-Delmotte, V., Schulz, M., Abe-Ouchi, A., Beer, J., Ganopolski, A., González-Rouco, J.F., Jansen, E., Lambeck, K., Luterbacher, J., Naish, T., Osborn, T., Otto-Bliesner, B., Quinn, T., Ramesh, R., Rojas, M., Shao, X., Timmermann, A.,:Information from Paleoclimate Archives, Climate Change 2013: The Physical Science Basis. Contribution of Working Group I to the Fifth Assessment Report of the Intergovernmental Panel on Climate Change, Cambridge University Press, Cambridge, United Kingdom and New York, NY, USA, chapter 5, 383?464, ISBN 978-1-107-66182-0, 10.1017/CBO9781107415324.013, 2013

Oleson, K. W., and Coauthors:Technical Description of version 4.0 of the Community Land Model (CLM),2010

Rhoades, A. M., Ullrich, P. A., Zarzycki, C. M., Johansen, H., Margulis, S. A. Morrison, H., Xu, Z., Collins, W. D.,: Sensitivity of Mountain Hydroclimate Simulations in Variable-Resolution CESM to Microphysics and Horizontal Resolution,Journal of Advances in Modeling Earth Systems,10,6,1357-1380,10.1029/2018MS001326,2018

Schmidt, G. A., and Coauthors, 2012: Climate forcing reconstructions for use in PMIP simulations of the last millennium (v1.1), Geosci. Model Dev.,5,185-191,2012.

---

## Author Response (AR1)

**Response to the reviewers' comments**

Camilo Melo-Aguilar [1,2], J. Fidel González-Rouco [1,2], Elena García-Bustamante [3], Jorge Navarro-Montesinos [3], and Norman Steinert [1,2]

[1]Universidad Complutense de Madrid, 28040 Madrid, Spain
[2]Instituto de Geociencias, Consejo Superior de Investigaciones Cientificas-Universidad Complutense de Madrid, 28040 Madrid, Spain
[3]Centro de Investigaciones Energéticas, Medioambientales y Tecnológicas (CIEMAT), 28040 Madrid, Spain
**Correspondence:** Camilo Melo-Aguilar (camelo@ucm.es)

The authors would like to thank the reviewers for their constructive suggestions and the time they devoted in reading and proof-reading the manuscript. We have tried to integrate all suggestions and think that the manuscript has improved with them. We do appreciate their contribution. The original manuscript with track changes is included so the specific modifications can be found in the text.

5 The next sections contain a detailed point by point response to the reviewers comments.

**1 Anonymous Referee 1**

*GENERAL COMMENTS:*

GC01: *REVIEWER'S COMMENT:*

10 *The manuscript presents a rather detailed investigation of the coupling between air surface temperature and ground temperature in a series of climate simulations over the past millennium with an Earth System model. The analysis is focused on two main aspects: their co-variability and inter-annual time scales and their relative trends over the period of anthropogenic warming. The main conclusions are that in general terms bot variables are strongly coupled, but with regional particularities. The main mechanism that modulate the coupling is snow cover. Considering the long-term*

15 *trends over the past century, varying snow cover due to warming causes a change of the coupling over time that should be taken into account when using ground temperature from boreholes to reconstruct past near surface-air temperature. Deforestation over the past century also affects the coupling between surface temperature and ground temperature. In my view the manuscript is very well written, and I have just a few minor suggestions that the authors may want to consider.*

AUTHORS' RESPONSE:

20 The authors welcome the positive perspective of the reviewer on the paper. We are grateful for the reviewer's comments. Please find below the comprehensive point-to-point response to your review. The original manuscript with track changes is included so the specific modifications can be found in the text.

GC02:  *REVIEWER'S COMMENT:*

*Sometimes, the text is a bit too wordy, with long paragraphs that pack a lot of information. This is admittedly a matter of style, but I guess that a typical reader would prefer some of the very long paragraphs broken up in smaller pieces, so that, after a first reading, the information can be located more rapidly if needed.*

5      AUTHORS' RESPONSE:

After a thorough revision of the text we have identified some paragraphs that have been broken up in smaller pieces or shortened. The main changes affect Sections 4 and 4.1. See pages 7, 8, 9, 11, 14 and 15 of the manuscript with track changes.

We hope that the reviewer finds the changes satisfactory and the quality of the manuscript improved.

10 GC03: *REVIEWER'S COMMENT:*

*The explanation of the negative correlation between air temperature and ground temperature in Tibet is intriguing, but I am not sure whether it is complete. The authors argue that the intermittent heat flux - modulated by seasonal snow cover - can cause a negative correlation between the time derivative of both temperatures, but does this translate in a negative correlation between the temperature anomalies themselves ? This may be true depending on the spectrum of the temperature variations.*

15      *If the spectrum is white, i.e. little multidecadal variability, I can see that this explanation can be correct. However, if the temperature variations are mostly decadal, changes in the time derivative of temperature may not be enough to change the sign of the temperature anomalies. All in all, I found this mechanism interesting, but I am not completely sure that it is the sole explanation. Unfortunately, I cannot suggest new ideas.*

AUTHORS' ANSWER:

20      Our explanation of the negative correlation between air and ground surface temperature is based on annual time series. Indeed, it refers specifically to anti-correlation between the anomalies themselves. Longer-term air and ground temperature variations, such as decadal or multi-decadal, are also anti-correlated, and intermittent heat flux modulated by seasonal snow cover is the dominant factor. A negative correlation between the time derivative of both temperatures is also present.

25      We have included a brief mention in the text stressing the fact that the anti-correlation refers specifically to the annual temperatures. See page 9, line 30 of the manuscript with track changes. Additionally, we have included in this document a figure depicting the influence of simulated snow cover and sensible heat fluxes on the anti-phase SAT-GST behavior at multi-decadal to centennial timescales (Fig. 1). Note that the negative correlation between SAT and GST is also evident for the 31-yr filter outputs. Additionally, snow cover and sensible heat fluxes also present this anti-correlation at the

30      multi-decadal timescales. Further, the anti-correlation is also notable at centennial timescales and trends at the end of the millennium. Also see response and changes related to reviewer #3 GC08.

*SPECIFIC COMMENTS:*

Other minor points:

SC01 *:* 'Improving our knowledge of the LM climate variability' LM has been defined in the abstract, but I think not in the main text.

   Answer: LM is now defined in the main text. Page 1, line 20 of the manuscript with track changes.

SC02 *:* The second, postulate is that variations in SAT propagate downward to the subsurface through conduction (Pollack et al., 1998; Smerdon et al., 2003, Delete comma after postulate

   Answer: The comma has been deleted. Page 2, line 22-23 of the manuscript with track changes.

SC03 *:* 'In the present work the LM refers to the period from 850 to 2005 CE while the periods from 850 to 1850 and 1851 to 2005 CE refer to pre-industrial and industrial, respectively. 2 m air temperature is used for SAT and the first land model level..' I my understanding, a sentence in English may not start with a number. Change to 'two-meter temperature..'

   Answer: The text has been changed accordingly. Page 6, line 8 of the manuscript with track changes.

SC04 *:* CESM-LME supports the assumption that SAT is tightly coupled with GST at global scales and above multi-decadal scales. May be change 'above' to 'longer than multidecadal temporal scales'

   Answer: This has been changed as suggested. Page 7, line 15 of the manuscript with track changes.

SC05 *:* 'the thermal difference between SAT and GST. In order to explore the influence of such processes on the global SAT-GST relationship this analysis is extended considering winter and summer seasons (DJF and JJA hereafter) independently (Fig. 1c,e).- Specify boreal summer and boreal winter

   Answer: This has been changed as suggested. Page 8, line 1-2 of the manuscript with track changes.

SC06 *:* The correlation maps (Fig. 2b) provide a similar pattern with high and significant values in most the globe (>0.8 in regions located below 45° north) and lower correlations over NH mid- and high-latitudes; especially over the east of Siberia. 'below' is here ambiguous (or even incorrect): does it mean between 45N and 90S ? or between 45N and the equator. Maybe simply say 'for northern latitudes smaller than 45 N

   Answer: The text has been changed to "(>0.8 in regions located between 45° N and 90° S). Page 9, line 8 of the manuscript with track changes.

**2 Anonymous Referee 2**

*GENERAL COMMENTS:*

GC01: *REVIEWER'S COMMENT:*

*This is overall a well written and interesting manuscript. It is a logical next step, given the work that some of these authors have done to test SATGST coupling and the implications for borehole paleoclimatology in LM simulations. It goes into detail to explain the various coupling behavior that is observed and has some important conclusions. I therefore think that it should ultimately be published, but I have significant concerns about some of the analysis as outlined below. A major revision is therefore necessary before the manuscript should be accepted for publication.*

AUTHORS' RESPONSE:

We appreciate the positive perspective on the manuscript as well as the comments and suggestions. We have considered each comment of the reviewer and we are sure that the new version of the manuscript have been improved.

Please find below the comprehensive point-to-point response. The original manuscript with track changes is included so the specific modifications can be found in the text

GC02: *REVIEWER'S COMMENT:*

*The biggest analysis issue that concerns me is the two-phase regression model. The authors do not give a reasonable justification for selecting this regression model and I think it has several significant impacts on their conclusions. In particular, the breakpoints influence their results in ways that can be hard to interpret. One way in which this is manifest is in their physical interpretations of the coupling influences. In Figure b, d, and f we are provided time series and regression results for SAT-GST and snow cover. The author's interpretation is that snow cover has influenced the coupling, but it appears that the breakpoint comes earlier in the SAT-GST time series than it does for snow cover. If we are to take the breakpoints as physically meaningful (and I think that should be done very cautiously), why should we conclude that snow cover is driving the decoupling when the major break in its trends comes after the breakpoint in SAT-GST? The answer is that we probably should not take the breakpoints very seriously and it is a shortcoming of the analysis that the regressions are confined to the two-phase model.*

AUTHORS' RESPONSE

We agree that improving the description of the two-phase regression and its use in this paper will be good for the text. Reviewer #3 has also shown some concern about this.

Changes in the text have been included to incorporate these improvements affecting the Methods and the Results section (3 and 4 respectively), modifying the test related to Figs. 1 and 9; Fig. 9 has also been changed (see description below and Fig. 2 of this document). See also page 6, line 33 and page 7, line 9-12 of the manuscript with track chances.

Indeed, the use of the two-phase regression has an impact on the text, to our opinion improving the understanding of

the results. The reason is that no condition is imposed on the time interval to evaluate trends and this allows for a better characterization of the SAT-GST responses to external forcings and feedbacks. This has been stated in the Methods (Sect. 3) part and also in the text related to Fig. 9 in Sect 4.1 (see page 14, line 4-10 of the manuscript with track changes). The modified text and Fig. 9 show better now that for many areas, changes in response to external forcing occur before 1850 CE, during the 17th and 18th centuries. We think that allowing the responses of SAT-GST to freely describe contributions in the timing and magnitude of the various single forcing factors to the ALL-F is interesting. We hope this is more clear now in the text.

The reviewer shows understandable concern for the physical interpretation of the timing of changes in snow cover in Fig. 1. We have made changes in the text describing this figure regarding the long-term trends at the end of the LM. We can recall that snow cover does play a role in multi-decadal/centennial in-phase (JJA) and anti-phase (annual and DJF) changes in pre-industrial times (see page 8, line 7-9 and 20-35 of the manuscript with track changes). Figure 1 shows two-phase trend results for SAT-GST and trends for snow cover time-locked to the changes in SAT-GST. The purpose of this is to show the same in-phase/anti-phase consistency described earlier for multi-decadal changes, acting here for long-term changes. We hope the text is more clear now regarding the interpretation of snow cover timing of changes and that these do not occur earlier or after those in SAT-GST. Further, in all examples provided elsewhere in Sect. 4.1 (e.g Figs. 11, 12, 13, 14) a physical interpretation is provided that supports the timing of two-phase regression changes.

GC03:  *REVIEWER'S COMMENT:*

*This is also a problem in the comparison of trends in Figure 9. The upper panels indicate that the break points vary significantly spatially, with some locations yielding breakpoints in the 1000-1200 period and many in the 1700-1800 period. All of these trends of course end at the beginning of the 21st century, meaning that the second-period trends displayed in the middle and bottom panels of Figure 9 are taken over vastly different time periods. It is very difficult to determine how to compare such trends, many of which will be impacted by strong 20th-century changes that are either much stronger because of a later breakpoint or much weaker because of an earlier breakpoint. One might justify this approach if the breakpoints could more reliably be interpreted, but in many cases, indeed as evidenced across the ensemble, there is a lot of variability in the determined timing of the breakpoint (look at Figure 15, for instance, and try to convince yourself that the break points in the upper panel for ALL-F or OZ/AER shouldn't be several centuries in either direction of where the breakpoints were determined to be). I therefore have a very hard time interpreting the magnitude of the trends and what they actually mean. As such, I think the two-phase regression model is both poorly justified and probably a very confusing interpretation of the data.*

AUTHORS' RESPONSE

We hope the changes in Fig. 9 make the interpretation to this part more clear now. The dates of change are now shown as an example for one member in each ensemble (Fig. 9a) and also exclusively for the significant areas in at least 80% of all members (Fig. 9b). It becomes obvious, as the reviewer indicates, that changes spread over the millennium, but

the important ones only take place in the last centuries, although not confined to post-1850. This is interesting and better highlights the physics of the time response of SAT-GST than if the trends were locked to the post-1850 period. The timing of the regional changes and its amplitude depends on the timing in the forcing changes and the influence of internal variability in each run. The fact that changes are consistent in 80% of the experiment members (Fig. 9b,d of the revised manuscript) indicates results are robust regarding the temporal and spatial variability of the changes described.

GC04:  *REVIEWER'S COMMENT:*

*Related to the above, the overall magnitude of the impact is hard to parse in the context of what the authors have done. They look specifically at the 1850-2005 period and note that the overall difference in SAT and GST trends looks to be about 0.05C. First of all, this should be couched in the context of the 20th-century trend estimate from the global borehole reconstruction of ~0.5C/century for the 20th century. If we take the model estimate at face value, the error introduced is therefore less 10%, given the authors' estimate is over a longer period of time (it should be reported in C/century for better comparison). But the other analyses complicate this number and I am not sure how to interpret them. The trends estimated in Figure 9 for SAT-GST are +/- 0.1C per DECADE, which is a much larger regional impact than the number determined for the global difference (I understand these regional impacts would be muted in a global average, but the numbers are complicated by my subsequent point). Part of the confusion is because of an apples-to-aardvarks comparison, namely the global trend difference is from 1850-2005, while the trends in Figure 9 are for widely different intervals determined by the breakpoint in the regression analysis. In some ways it would be much cleaner if the authors determined trends over set centuries to make sure the comparisons were consistent in time and to make the comparisons to the actual borehole reconstructions simpler. Such an analysis would have its own shortcomings, but as things stand the presentation is ambiguous on the most important point of the paper, namely if and how big a bias might be present in the borehole reconstructions based on the authors' analysis.*

AUTHORS' RESPONSE

The use of a fixed time period, as for instance from 1850 to 2005, to determine the trends as the reviewer suggests is indeed an interesting approach that we also considered and is adopted in Fig. 8. On the other hand, allowing for the trends to optimally show when changes related to each forcing occur (Fig. 9), provides arguably more complete information and of interest for a broader community. As mentioned above, it also shows that the dates and magnitude of changes are spatially and temporally consistent; also that changes pre-date 1850. We are not aware of any text, having shown the temperature changes in response to each forcing in the context of the last millennium, describing the spatial dependence of the response without imposing time bounds. Indeed, the text states that implications for borehole reconstructions are marginal at global scales. Some editing has been done in this direction in page 8, line 26-35 of the manuscript with track changes. This is also stated in the Conclusions in page 20 lines 25-35. However, the influences at subcontinental scales could be relevant. A more quantitative evaluation requieres a more detailed implementation of borehole inversion methods (González-Rouco et al. , 2006, 2009). This will be done elsewhere.

GC05: *REVIEWER'S COMMENT:*

*Another issue related to the interpretation of the authors' results in the context of the borehole reconstructions is the actual spatial sampling of the borehole database and how it may actually be impacted by the biases the authors are reporting. This is particularly the case regarding snow cover and the other cryogenic effects that the authors report for the high latitudes. While these impacts are reasonably described and there are clear differences between GST and SAT particularly in the high northern latitudes, there simply aren?t that many boreholes used for the global reconstructions above about 55-60 degrees N latitude.*

*While the biases are certainly a concern and relevant to any regional studies of high-latitude boreholes, it is not clear that the regionally very large impacts play much of a role in the existing NH and global borehole reconstructions. The authors could do a lot to quantify the effect by performing an analysis that restricts the assessment to locations where boreholes exist in the reconstruction database. I would advocate for that, but it would also add to what is already a long and detailed paper. It would suffice to simply mention this issue and \*perhaps\* provide at least some numbers on the impact of the hemispheric or global estimates if, for instance, latitudes north of 60 degrees were excluded.*

AUTHORS' RESPONSE:

We have been considering the analysis of the impacts that this kind of long-term SAT-GST decoupling processes may have on the actual spatial distribution of the borehole datase. However, this goes beyond the scope of the present work, as indicated in the response to GC04. We are currently working on a paper that explores in detail the impacts of decoupling in the context of the actual borehole distribution may have on borehole reconstructions at different spatial scales. Thus, the specific numbers will be addressed in the subsequent work.

**SPECIFIC COMMENTS:**

The paper is generally well written, but there are language issues scattered throughout. I have noted some of those below, but the paper would be improved by a thorough evaluation of typos and language choices.

SC01 *Pg. 3, ln 9:* has been supported by observations...

Answer: This has been corrected. Page 3, line 9 of the manuscript with track changes.

SC02 *Pg 4, ln 2:* PPEs experiments is redundant

Answer: The text has been deleted. Page 4, line 2 of the manuscript with track changes.

SC03 *Pg 6, ln 1:* It composes a total...

Answer: The text has been changed as suggested. Page 6, line 1 of the manuscript with track changes.

SC04 *Pg 6, ln 12:* covariance during...

Answer: This has been corrected. Page 6, line 12-13 of the manuscript with track changes.

SC05 *Pg 6, ln 16:* may help characterize a potential...

Answer: The text has been changed as suggested. Page 6, line 16 of the manuscript with track changes

SC06 *Pg 6, ln 28:* associations

Answer: The expression refers to a singular form therefore it is not changed in the document

SC07 *Pg 7, ln 4:* I don't think that the OZ/AER (ozone/aerosol) abbreviation has been defined in text.

Answer: It is defined in Table 2, which is mentioned in Section 2 specifically in page 6, line 3 of the manuscript with track changes.

SC08 *Pg 7, ln 5:* allows identification

Answer: The text has been corrected accordingly. Page 7, line 5 of the manuscript with track changes.

SC09 *Pg 7, ln 7:* through industrial times

Answer:This has been corrected. Page 7, line 8 of the manuscript with track changes.

SC10 *Pg 7, ln 3:* I don't think the subscript notation for the ensemble member or the layer depth has been explained anywhere. It should be defined for completeness.

Answer: The subscript notation for the ensemble member and for the layer depth is explained in Table 3 and Table 1 respectively (Section 2). Additionally, the manuscript with track changes (page 6, line 9-10) includes now further comments on this.

SC11 *Pg 7, ln 20:* While the correlations with GST and the lower depths are perhaps useful to explore, it should be explained that because of the phase shift correlations will diminish even if the system is purely conductive. All borehole inversions take into account the phase shift and this kind of time series comparison is a bit misleading. The authors should clearly point this out if they are going to leave this analysis in the manuscript. For what it is worth, some of the length and detail of the manuscript would be improved by removing these specific analyses because I do not think they add much.

Answer: We agree with the reviewer and actually nor the text nor the figures contradict this. We have included a brief note stressing this issue. See page 7, line 23 and 26-27 of the manuscript with track changes.
We think that the analysis of the correlation with GST and the lower $ST_{L8}$ deserves being included since it gives some relevant information about the propagation of the temperature signal with depth. On the one hand, the time series allow for illustrating that the CESM-LME represents the characteristics of the conductive regime, as amplitude attenuation and phase shift of the temperature signal with depth. On the other hand, the spatial analysis presented in Fig. 6 permits illustrating the influence of non-conductive processes on the subsurface heat transfer over certain areas and the amplitude attenuation with depth of the temperature signal at annual time scales that may be of interest for some readers.

SC12 *Pg 7, ln 26:* very cold air temperatures

Answer: The text has been changed. Page 7, line 31 of the manuscript with track changes.

SC13 *Pg 8, ln 30:* consider "In contrast" instead of "On the contrary"

    Answer: This has been replaced. Page 9, line 13 of the manuscript with track changes.

SC14 *Pg 9, ln 28:* relationship at relatively

    Answer: The text has been changed. Page 10, line 14 of the manuscript with track changes.

SC15 *Pg 10, ln 3:* during JJA consistent with

    Answer: The text has been changed. Page 10, line 24 of the manuscript with track changes.

SC16 *Pg 10, ln 8:* high rates

    Answer: This has been corrected. Page 10, line 33 of the manuscript with track changes.

SC17 *Pg 10, ln 24:* such partitioning determines

    Answer: It has been changed. Page 11, line 11 of the manuscript with track changes.

SC18 *Pg 10, ln 27:* because the water, relative to the continental air, is warmer in winter.

    Answer: We have included some changes affecting this sentence. Page 11, line 14-15 of the manuscript with track changes.

SC19 *Pg 10, ln 33:* The paragraph that ends here is an example of where things get a little too detailed. I think that this paragraph in particular could be scaled down a bit.

    Answer: We have done a thorough revision of the text we have identified some paragraphs, including this specific one, that have been broken up in smaller pieces or shortened. The main changes affect Sections 4 and 4.1. See pages 7, 8, 9, 11, 14 and 15 of the manuscript with track changes.

SC20 *Pg 11, ln 14:* during these months

    Answer: The text has been corrected. Page 12, line 2-3 of the manuscript with track changes.

SC21 *Pg 12, lns 13-14:* It is relevant that other GCMs may suggest different levels of impacts, but this seems a little strange in the context of the real world. The real question is how well the GCMs represent the real-world impacts. It would be useful here to make a statement along those lines, in connection to what consistency or disagreements across models might imply for the real world.

    Answer: Regarding how well CGMs represent the real world, this is a broad issue (Flato et al. , 2013). This work describes the most relevant physical mechanisms and feedbacks that may affect SAT-GST coupling in different parts of the world, many of which have not yet been measured systematically, and how these processes interact with long-term changes in external forcing factors. For this purpose we have used the largest and only ensemble of model simulations including several all-forcing and single forcing realizations. We would expect that largest ensembles with a more systematic sampling of forcings and processes will be available in CMIP6/PMIP4 (Eyring et al. , 2017; Jungclaus et al. ,

2017). This will allow for exercises exploring better the sensitivity to using different models.

We have included a discussion on this issue in the conclusions part of the revised manuscript. See page 20 line 25-35 of the manuscript with track changes.

SC22 *Pg 12, ln 16:* trends of both temperatures independently

Answer: The text has been corrected. Page 13, line 6 of the manuscript with track changes

SC23 *Pg 13, ln 1:* Why not report pattern correlations in this discussion? A quantification of the spatial similarity would seem helpful.

Answer: The pattern correlation has been included. See page 13, line 26-27 of the manuscript with track changes.

SC24 *Pg 15, ln 15:* the driving mechanisms

Answer: This has been corrected. Page 16, line 18 of the manuscript with track changes.

SC25 *Pg 15, ln 26:* The authors say nothing about simulated increases in LAI here. Instead of transitions to different plant functional types, it is also possible that increases in CO2 during the 20th century drive LAI increases in the CESM that in turn impact evapotranspiration. This has been shown for the model projections and also may play a role in the historical portion of the runs that the authors are analyzing. See the following paper for some background: https://agupubs.onlinelibrary.wiley.com/doi/abs/10.1002/2018GL077051

Answer: It is indeed very interesting that increases in LAI, as a response to enriched $CO_2$ environment, lead to increases in evapotranspiration, and possibly to long-term variations in the SAT-GST relationship. Actually, the Community land model version 4 (CLM4), the land component included in the CESM, incorporates a Carbon-Nitrogen (CN) module (Oleson et al. , 2010). This module permits the allocation of the available carbon to new plant growth. In fact, due to the increase in CO2 concentration during the industrial period, there is an increase in gross primary productivity in the CESM-LME simulations that consider the GHG forcing. The CN module also calculates the $CO_2$ atmospheric concentrations originating in land use changes and other sources. However, it is radiatively inactive since the radiative forcing is prescribed (Lehner et al. , 2015).

We have identified that there are indeed some areas where increases in LAI during the industrial period lead to variations in the long-term SAT-GST relationship in the CESM-LME. Nonetheless, the impact of this mechanism is relatively small compared to others described in the manuscript. We have included a brief mention of this issue. See page 18, line 17-19 of the manuscript with track changes.

SC26 *Pg 15, ln 32:* stable state

Answer: This has been corrected. Page 17, line 1 of the manuscript with track changes

**3 Anonymous Referee 3**

GENERAL COMMENTS:

GC01: *REVIEWER'S COMMENT:*

*The paper examines the relationship between surface air temperature and ground surface temperature within a set of model simulations. The idea is to use the SAT-GST correlation as a metric to examine – or as a proxy, for the ground surface energy balance. This is done with the CESM-LME model ensemble that includes a set of experiments with all forcing as well as simulations with individual forcing. It is expected that comparison of the resulting simulations would yield information regarding the importance of each of these forcing on the SAT-GST relation over spatial scales and its temporal evolution. The paper briefly mentions that this may be important for borehole climatology based climate reconstructions.*

*This is a generally well-written papers, but it would benefit from a full language revision.*

*I have several minor clarifications that I believe need to be implemented for completeness before the paper is published.*

AUTHORS' RESPONSE:

We appreciate the positive perspective on the manuscript as well as the comments and suggestions. We have considered each comment of the reviewer and we are sure that the new version of the manuscript has been improved.

Please find below the comprehensive point-to-point response. The original manuscript with track changes is included so the specific modifications can be found in the text

GC02: *REVIEWER'S COMMENT:*

*What do "regional" and "local scale" mean in the context of this model resolution?*

AUTHORS' RESPONSE

Considering the horizontal resolution of the CESM-LME ~2°, in this work local scale refers to the scale of resolution a single or a few grid points while for the regional scale the range between $10^4$ to $10^7$ km$^2$ can be considered as defined in (Giorgi et al. , 2001). From this perspective it is really broad and that range can refer in a general sense to sub-continental structures that can be as small as the province of Madrid and as large as the U.S. We will consider the use of this concepts and make sure that it is senseful in the revised text.

GC03: *REVIEWER'S COMMENT:*

*How is the SAT defined within the CESM? Is it always the surface air temperature at a height of 2 m above the bare ground? Is it defined differently over a (say) forested area? If so, what could be the potential problems for the SAT-GST correlation?*

AUTHORS' RESPONSE

The variable we used for SAT is the temperature near the surface or 2-m air temperature identified by the model output "TREFHT".

TREFHT is calculated by interpolation between the surface temperature and the lowest atmosphere model level temperature. The surface temperature depends on the energy fluxes at the surface. For non-vegetated areas, the surface temperature is obtained from the energy balance at the ground surface. For vegetated surfaces, the energy fluxes are partitioned into vegetation and ground fluxes (Oleson et al. , 2010). Therefore, the surface temperature is the result of the balance between both the vegetation and the ground fluxes.

Since in both cases the energy fluxes from the different surface components are considered in the calculations of the surface temperature, and consequently on the 2-m air temperature, there should not be "potential problems" for the SAT-GST correlation.

From the surrogate reality of the GCMs simulated air temperature represents the best available alternative for exploring the air and ground temperature relationship. Previous works of this type have also used 2-m air temperature from different GCMs for investigating the covariance structure between the air and ground temperatures (e.g. González-Rouco et al. , 2009; García-García et al. , 2009).

GC04: *REVIEWER'S COMMENT:*
*Could the author comment on the role played by evapotranspiration on the SAT-GST metric?*

AUTHORS' RESPONSE

Evapotranspiration may play an important role on heat transfer from the surface to the atmosphere. It is involved in the energy balance at the surface. However, evapotranspiration does not necessarily influence the SAT-GST metric by itself. The SAT-GST long-term relationship may be affected if the means by which heat is transfered from the surface to the atmosphere experiences a long-term variations. There are a couple of examples in the document describing how long-term variations in evapotranspiration, as a response to changes in the surface characteristics, lead to long-term changes in the SAT-GST metric. For instance in Fig. 12 of the submitted manuscript it is presented how a significant increase in evapotranspiration impacts the SAT-GST relationship.

On the contrary, if the energy fluxes at the surface are relatively stable in time, the SAT-GST metric would remain also stable.

GC05: *REVIEWER'S COMMENT:*
*What is the justification for using the two-phase regression model?*

AUTHORS' RESPONSE

This comment is also raised by anonymous referee #2. Please for the answer to this comment refer to GC02 of reviewer#2

GC06: *REVIEWER'S COMMENT:*
*Why do authors use the "change points" obtained from the SAT-GST regression for the analogous analysis for snow*

*cover? Shouldn't these snow cover "change points" be determined independently in order to see whether Is a relationship exists?*

AUTHORS' RESPONSE

Please for the answer to this comment refer to GC02 of reviewer#2

5 GC07: *REVIEWER'S COMMENT:*

*DJF and JJA refer to northern Hemisphere seasons, but are used in the global context. There is some discussion later in the text to acknowledge these restrictions, but they should be given up front or simply restrict the analysis to the northern hemisphere; likely there would be no difference in the results.*

AUTHORS' RESPONSE

10 We have included a explanation indicating that DJF and JJA refer specifically to the boreal winter and summer respectively. Page 8, line 1-2 of the manuscript with track changes.

GC08: *REVIEWER'S COMMENT:*

*Regarding the discussion about the Tibetan Plateau (Fig 4) , could the authors discuss whether the CESM resolution can account for the topographic variations and whether such high elevation variability could introduce unforeseen effects*
15 *on SAT-GST?*

AUTHORS' RESPONSE

Horizontal model resolution does not seem to be an issue since a higher resolution version of the model (CCSM4) produces a similar behavior and other model of similar resolution do not show it (García-García et al. , 2009). This comment has been included in page 10, line 8-11

20 GC09: *REVIEWER'S COMMENT:*

*In Figure 9, are all trends for the"post-change" interval? If so, why are the values given in K/decade? Most change points are post 1850; there should be at least a century post change. i.e. could you give the changes in k/century?*

AUTHORS' RESPONSE

The trends are indicated in C/century in the Manuscript with track changes.

25 GC010: *REVIEWER'S COMMENT:*

*I am not convinced that the results from this work necessarily imply that borehole climatology reconstructions would have to be revised to account for SAT-GST uncoupling. The effects are small and given the uncertainties inherent to the estimates of the quasi-steady state of the thermal regime of the ground, these additional errors are likely very small. Perhaps a simple theoretical experiment with +/- the LUC uncertainty imposed on a typical borehole temperature profile*
30 *and its resulting inversion may illustrate the effect well, if any.*

AUTHORS' RESPONSE

We are currently working on a paper that aims to explore in detail the impacts in the context of the actual borehole distribution may have on borehole reconstructions at different spatial scales. Thus, the specific numbers will be addressed in the subsequent work. See also response to GC04 of reviewer #2.

[Figure]

**Figure 1.** LM evolution of surface air temperature (SAT), ground surface temperature (GST), snow cover fraction (Snow) and sensible heat flux (SHFL) 
[revised manuscript text omitted]